# LEARNING PERSONALIZED DRIVING STYLES VIA REINFORCEMENT LEARNING FROM HUMAN FEEDBACK

## ABSTRACT

Generating human-like and adaptive trajectories is essential for autonomous driving in dynamic environments. While generative models have shown promise in synthesizing feasible trajectories, they often fail to capture the nuanced variability of personalized driving styles due to dataset biases and distributional shifts. To address this, we introduce TrajHF, a human feedback-driven finetuning framework for generative trajectory models, designed to align motion planning with diverse driving styles. TrajHF incorporates multi-conditional denoiser and reinforcement learning with human feedback to refine multi-modal trajectory generation beyond conventional imitation learning. This enables better alignment with human driving preferences while maintaining safety and feasibility constraints. TrajHF achieves performance competitive to the state-of-the-art on NavSim benchmark. TrajHF sets a new paradigm for personalized and adaptable trajectory generation in autonomous driving.

## 1 INTRODUCTION

In autonomous driving, trajectory planning is responsible for generating safe, feasible, and human-like motions in complex and dynamic traffic environments. Heuristics-based learning methods (Zhao et al., 2021; Gu et al., 2021; Xing et al., 2025) perform well in structured scenarios but struggle to generalize across diverse driving conditions. Imitation learning, most prominently generative models (Jiang et al., 2023b; 2024; Ngiam et al., 2021; Sun et al., 2023), provides a data-driven alternative. They synthesize diverse human-like trajectories, but do not explicitly capture the nuanced preferences that characterize human driving behaviors, such as individual tendencies, regional norms, and environmental factors.

Learning from demonstrations is fundamentally constrained by dataset bias. Behavior cloning only captures an average driving behavior, failing to represent the full spectrum of maneuvers. Generative imitation learning, on the other hand, learns real data distributions in the dataset, but is usually dominated by frequent modes, and hardly performs the best behavior unless sampling is conducted in an exhaustive manner. This misalignment degrades model performance in scenarios requiring deviations from frequent demonstrations, such as aggressive adaptations in complex traffic interactions, as shown in Fig. 1. Furthermore, human driving styles are influenced not only by kinematic constraints but also by higher-level cognitive and social factors, such as risk tolerance, interactions with other road users, and situational awareness. Existing methods struggle to encode these latent factors, leading to planned motions that, while technically feasible, may feel unnatural, overly conservative, or unpredictably assertive. This further reduces user trust and acceptability, underscoring the need for trajectory generation methods that align with diverse human driving behaviors.

To address this challenge, we propose TrajHF, a novel framework to finetune generative trajectory models using human feedback to align trajectory generation with diverse personalized driving styles. Human feedback, in the form of comparative trajectory rankings or explicit annotations, provides a rich supervisory signal that extends beyond conventional imitation learning paradigms. By integrating this feedback into the finetuning process, our approach enables generative models to capture human-preferred driving styles while maintaining safety and feasibility constraints.

We systematically investigate methodologies for incorporating human feedback into trajectory model finetuning. We propose a novel Multi-Conditioned Denoiser (MCD) Transformer network to gen-

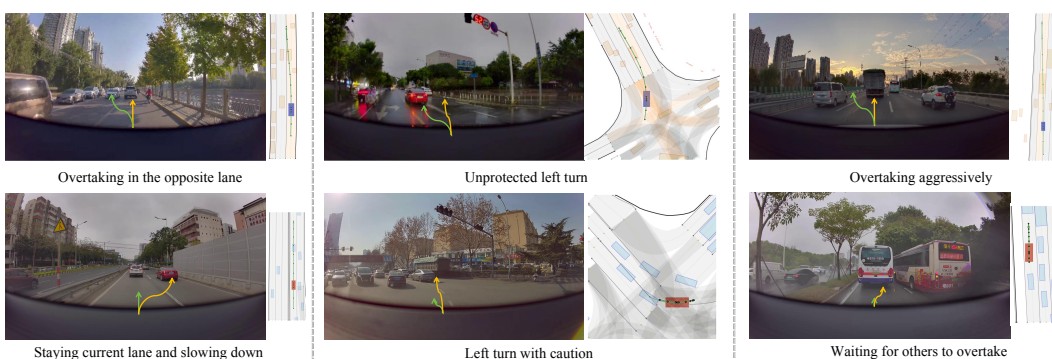

Figure 1: **Illustration of human preference problem in autonomous driving problem.** The first line is aggressive driving style examples and the second line is defensive style. Yellow lines indicate "normal" planning trajectories and green lines represent the stylized ("aggressive" or "defensive") human preferred trajectories.

erate multi-modal trajectory planning. Our design of conditioning multi-modal information gains significant improvement of trajectory generation quality, reaching performance competitive to the state-of-the-art on NavSim (Dauner et al., 2024) benchmark with uniform sampling and optimization, without any explicit guidance like anchors. Meanwhile, we explore reinforcement learning with human feedback and leverage group computation gathering rewards from each mode of generated trajectories to iteratively refine the generative trajectory distribution. Experimental results demonstrate that our method improves alignment with distinct driving styles in human preference data ("aggressive" or "defensive", respectively) while preserving critical motion planning constraints. Using human feedback as an auxiliary learning signal, TrajHF offers a more personalized and adaptable trajectory generation paradigm, paving the way for autonomous vehicles that better reflect human driving behavior while ensuring safety and efficiency.

The primary contributions of this paper are as follows:

1. We systematically investigate the trajectory distribution shift problem between the average behaviors and the personalized driving styles.

2. We propose a human feedback-driven finetuning framework for generative trajectory models, enabling alignment with diverse human driving preferences.

3. Comprehensive experimental evaluations demonstrate the effectiveness of TrajHF in generating human-aligned autonomous driving trajectories across diverse scenarios.

## 2 RELATED WORK

### 2.1 END-TO-END TRAJECTORY PLANNING

Imitation learning (IL) has been widely adopted for end-to-end trajectory planning from expert demonstrations (Codevilla et al., 2018). UniAD (Hu et al., 2023) and ViP3D (Gu et al., 2023) leverage BEV inputs and transformer architectures to build planning-centric frameworks. VAD (Jiang et al., 2023a) and VADv2 (Chen et al., 2024) discretize trajectory space to transform regression into classification via trajectory vocabularies. NavSim (Dauner et al., 2024) introduces a benchmark focused on interaction-rich driving scenarios. Transfuser (Chitta et al., 2022) fuses lidar and image features, and FusionAD (Ye et al., 2023) propagates this fusion to downstream modules. Hydra-MDP (Li et al., 2024) distills multi-source knowledge through trajectory vocabulary. In contrast, our diffusion-based planner directly generates continuous, multi-modal trajectories without anchors or discretized spaces, supporting robust planning in complex scenes.

### 2.2 GENERATIVE TRAJECTORY MODELS

Generative models provide a principled way to capture multi-modal driving behaviors. Earlier works use GANs (Gupta et al., 2018; Fang et al., 2022) or VAEs (Xu et al., 2022; De Miguel et al., 2022),

while recent advances rely on autoregressive (Sun et al., 2023) and diffusion models (Jiang et al., 2023b; 2024). In end-to-end settings, VBD (Huang et al., 2024) combines denoising with behavior prediction, DiffusionDrive (Liao et al., 2024b) reduces computation via truncated inference, and GoalFlow (Xing et al., 2025) integrates goal-conditioned flow matching. However, these models prioritize feasibility over personalization, lacking mechanisms to capture the diversity of human driving preferences, which is addressed in our proposed framework TrajHF.

## 2.3 FINETUNING WITH HUMAN FEEDBACK

Reinforcement Learning from Human Feedback (RLHF) (Christiano et al., 2017) integrates subjective preferences via reinforcement learning and has shown success across domains (Ouyang et al., 2022; Yu et al., 2024; Zhang et al., 2024; Shao et al., 2024). Instead of relying on dense annotations, it optimizes models toward human-preferred distributions from limited feedback. Recent work applies RLHF to diffusion models (Wallace et al., 2024) using preference optimization without explicit rewards. In driving tasks, Wang et al. (2024) train reward models on trajectory preferences, and Sun et al. (2024) use coarse-grained safety preference data with KL penalties. We adopt GRPO (Guo et al., 2025), a critic-free method for efficient reward-based finetuning, and combine it with behavior cloning (Ross & Bagnell, 2010) to retain model capabilities.

## 3 PROBLEM DEFINITION

Let the state space of an autonomous vehicle be denoted as $S$, where each state $s_l \in S$ indicates the kinematic properties (e.g., position and heading) of the agent at timestamp $l$. A trajectory $x$ is defined as a sequence of states,

$$x = \{s_0, s_1, \ldots, s_L\} \tag{1}$$

where $L$ is the trajectory length. The expert trajectory distribution $P_{\text{data}}(x)$ reflects human driving behaviors in the training dataset, whereas $P_\theta(x)$ represents the distribution induced by a generative trajectory model parameterized by $\theta$. In a typical imitation learning setup, the goal is to minimize the discrepancy between these distributions, which can be formalized as the Kullback-Leibler (KL) divergence:

$$D_{KL}(P_{\text{data}} \| P_\theta) = \sum_x P_{\text{data}}(x) \log \frac{P_{\text{data}}(x)}{P_\theta(x)} \tag{2}$$

We believe that personalized driving styles can be depicted by human preferences when making decisions in complex driving scenarios. In this work, we deem styles and preferences as two equivalent concepts. Human driving preferences can be characterized by a latent distribution $P_{\text{pref}}(x)$, with each driver or style corresponding to a distinct sub-distribution. Since expert demonstrations typically represent an aggregated distribution rather than explicitly capturing individual variations, the actual target distribution can be seen as a mixture of preference distributions,

$$P_{\text{data}}(x) = \sum_i w_i P_{\text{pref}}^i(x) \tag{3}$$

where $P_{\text{pref}}^i(x)$ represents the trajectory distribution of the $i$-th driving style, and $w_i$ is its mixture weight. Training solely on $P_{\text{data}}(x)$ may cause the generative model to miss the underlying multi-modal structure, leading to suboptimal performance on trajectories outside the dominant mode of the dataset. To align the generated trajectory distribution $P_\theta(x)$ with diverse human preferences, we introduce a human feedback finetuning approach that incorporates preference annotations into the training process. Specifically, given a set of human-ranked trajectories $HF = \{x_1, x_2, \ldots, x_N\}$ annotated by human, we define a preference-based reward function $R(x)$ such that,

$$P_{HF} = \arg\max_P \mathbb{E}_P[R], \tag{4}$$

where $P_{HF}(x)$ represents the refined trajectory distribution incorporating human feedback. This enables the model to minimize the divergence $D_{KL}(P_{HF} \| P_\theta)$, thereby reducing the discrepancy between the learned trajectory distribution and the human-preferred trajectory distribution.

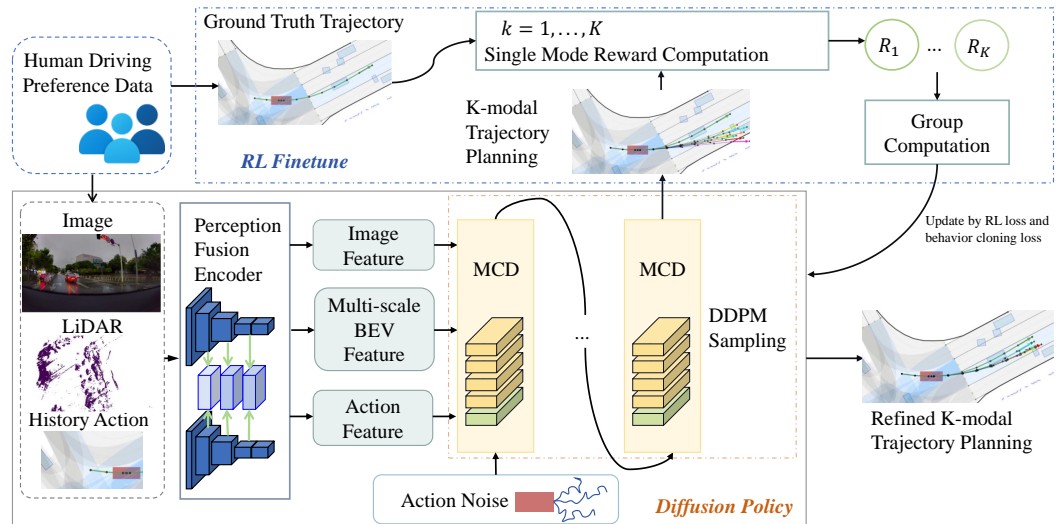

Figure 2: **Overview of TrajHF system.** After pretraining, our diffusion policy generates $K$-modal trajectories conditioned on multi-modal inputs from human driving preference data via the Multi-Conditional Denoiser (MCD). Each trajectory mode is compared with the stylized ground truth trajectory to compute a reward. The $K$ rewards by modes are fed into group computation and used to finetune the diffusion policy through a combination of RL loss and behavior cloning loss. The system refines the multi-modal autonomous driving planning to align with the driving style knowledge in the preference data.

# 4 METHOD

The overall architecture of our system is illustrated in Fig. 2. It includes two key components: diffusion policy for multi-modal trajectory generation and reinforcement learning (RL) finetuning which aligns the model with the human preference data.

## 4.1 GENERATIVE TRAJECTORY MODEL

**Diffusion Model Preliminaries** Diffusion models learn to represent the trajectory distribution $P_\theta(x)$ by minimizing the $KL$-divergence defined in Eq. 2. We adopt Denoising Diffusion Probabilistic Models (DDPM) (Ho et al., 2020) as the foundational framework for the denoising process. DDPM establishes a reverse denoising process that transitions from Gaussian noise $x_T$ to the noise-free state $x_0$, governed by the following transition equation:

$$x_{t-1} = \frac{1}{\sqrt{\alpha_t}} \left( x_t - \frac{1-\alpha_t}{\sqrt{1-\bar{\alpha}_t}} \epsilon_\theta(x_t, t) \right) + \sigma_t z \tag{5}$$

where $z \sim \mathcal{N}(\mathbf{0}, \mathbf{I})$. $t \in \{1, \ldots, T\}$ denotes the noise level, and $\epsilon_\theta$ represents the noise prediction network with learnable parameters $\theta$. The Gaussian noise added to the data $x$ is according to a variance schedule $\beta_t$. Let $\alpha_t = 1 - \beta_t$, $\bar{\alpha}_t = \prod_{i=1}^{t} \alpha_i$ and $\sigma_t^2 = \beta_t$. The denoising model takes the current noise level and perceptual conditions as input to estimate the noise. We formalize the conditional distribution to be predicted as $P_\theta(x|o)$, where $o$ is the observation conditions input, including front-view images, LiDAR sensors, historical actions and ego states. The optimization objective is formulated as a gradient descent step on:

$$\nabla_\theta \|\epsilon - \epsilon_\theta(\sqrt{\bar{\alpha}_t}x_0 + \sqrt{1-\bar{\alpha}_t}\epsilon, t, o)\|^2 \tag{6}$$

where $\epsilon$ is randomly sampled from $\mathcal{N}(\mathbf{0}, \mathbf{I})$.

**Multi-Conditional Denoiser** To solve the complex autonomous driving problem, we design a denoising transformer to generate trajectories conditioned on multi-modal perceptions, including front-view images, LiDAR data and historical states, as shown in Fig. 3. Before introducing Gaussian noise into the model input, the ego-centric trajectory is initially projected into the action space to mitigate heteroscedasticity along the trajectory timeline. The projection is expressed as $\hat{x}_l = s_l - s_{l-1}$

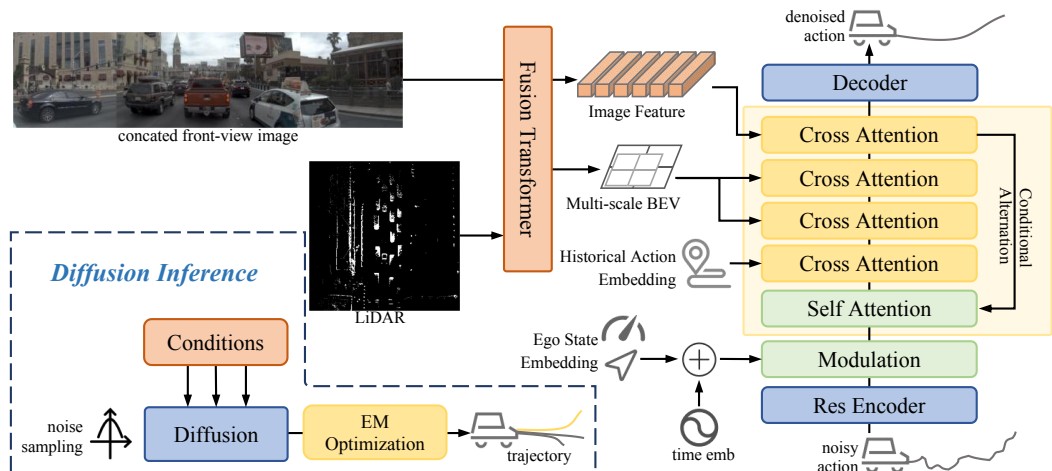

Figure 3: **The detailed architecture of multi-conditional denoiser.** The multi-modal perceptual input (image, LiDAR and actions) is encoded to different features which serve as the conditions of our transformer denoiser. During inference, multiple uniform sampling leads to multi-modal trajectory planning, which is then optimized to the final result by an EM algorithm.

and $s_0 = \mathbf{0}$ is the initial state, where $\hat{x}_l$ represents the agent's action at timestep $l$, and $l$ ranges from 1 to $L$. This mapping ensures a reversible relationship between the two spaces, allowing the trajectory to be readily derived by accumulating actions. The subsequent noisy action is encoded by a MLP with residual, followed by modulation with state and time embeddings. Our model integrates multiple contextual conditions to guide the denoising process. Inspired by Transfuser (Chitta et al., 2022), the concatenated front-view image and LiDAR are primarily encoded by their backbones, respectively, interleaved with fusion transformers to provide visual and BEV conditions. The intermediate fused feature maps are further upscaled to serve as a higher-resolution BEV condition. These conditions interact with the noisy action through cross attention blocks over multiple iterations, ultimately producing the predicted noise, which is then utilized in the diffusion process. During inference, the model generates a pool of candidates. An EM-algorithm based post-processing methods inspired by Varadarajan et al. (2021) is used to obtain the final planning trajectory out of inference samples, detailed in Appendix B.1. Notably, this approach is entirely **free** from the use of anchors or vocabulary-driven mechanisms.

## 4.2 DRIVING PREFERENCES ALIGNMENT

To acquire quantified human driving preference, we construct dataset and train a reward model $r_\phi$ to evaluate the generated trajectory styles. Based on this model, we develop the Group Reward Policy Optimization for Diffusion Perference finetuning (DPGRPO) algorithm, ensuring that the generated trajectories are aligned with human preference while regressing toward practical form.

**Training Reward Model with Semi-synthetic Data** We construct a human driving preference dataset $\mathcal{D}_p = \{(o^{(j)}; \mathbf{x}_c^{(j)})\}_{j=1}^n$ by mining critical scenarios from takeover data and identifying distinct driving styles. The $o$ represents an observation in a driving scenario and the $\mathbf{x}_c$ represents the chosen trajectory aligned with human preference. Based on this dataset, we develop a semi-synthetic dataset $\mathcal{D}_R = \{(o^{(j)}; \mathbf{x}_c^{(j)}, \mathbf{x}_i^{(j)})\}_{j=1}^m$ for reward model training, where the ignored trajectory $\mathbf{x}_i$ is synthesized from our pretrained model and represents normal trajectory. The details of preference data collection is presented in Appendix A.1 and the details of semi-synthetic dataset construction are presented in Appendix A.2.

We employ the same denoising transformer structure described in Section 4.1 as the encoder of the reward model $r_\phi$, processing generated trajectory $\mathbf{x}$ and observation information $o$ including camera feature, lidar feature and past trajectory. For the decoder, we directly convert this encoded representation into scalar rewards using a multilayer perceptron.

---

**Algorithm 1** DPGRPO

---

**Input:** diffusion policy $\pi_\theta$, dataset of the target driving style $\mathcal{D}_p$, group size $K$.
**Output:** finetuned policy $\pi_\theta$.
 1: Initialize reference policy $\pi_\theta^{\text{ref}} \leftarrow \pi_\theta$
 2: **while not** converged **do**
 3:     Sample $(o, z) \sim p(o) \times \mathcal{N}(\mathbf{0}, \mathbf{I})$
 4:     Sample a trajectory group $\mathcal{G} = \{\tau^{(k)}\}_{k=1}^K \sim \mathcal{P}_\theta(o, z)$
 5:     **for** $k = 1, \ldots, K$ **do**
 6:         Compute return $r_k \leftarrow r_\phi\big(o, \mathbf{x}_0^{(k)}\big)$
 7:     Compute estimated group relative advantages $\hat{A}_{\text{gr}}^k$ via Eq. 8
 8:     Compute RL loss $\mathcal{L}_{\text{RL}}^\theta$ via Eq. 9
 9:     Sample a reference trajectory $\tau^{\text{ref}} \sim \mathcal{P}_{\text{ref}}$
10:     Compute BC loss $\mathcal{L}_{\text{BC}}^\theta$ via Eq. 10
11:     Update $\pi_\theta$ by minimizing the combined loss $\mathcal{L}^\theta$ of Eq. 11
12: Apply short supervised refresh with $\mathcal{D}_p$
13: **return** converged policy $\pi_\theta$

---

We implement a Bradley-Terry ranking loss (Bradley & Terry, 1952) with margin as our reward model loss function, defined as:

$$\mathcal{L}^\phi = \mathbb{E}_{(o; \mathbf{x}_c, \mathbf{x}_i) \sim \mathcal{D}_R}\Big[ -\log \sigma\big(r_\phi(o, \mathbf{x}_c) - r_\phi(o, \mathbf{x}_i)\big) + \big[m - (r_\phi(o, \mathbf{x}_c) - r_\phi(o, \mathbf{x}_i)\big]_+ \Big], \quad (7)$$

where $\sigma$ represents the sigmoid function, $[\cdot]_+$ is equivalent to $\max(0, \cdot)$ and $m$ is margin constant.

**Finetuning Trajectory Model with Preference RL** We formulate the diffusion denoising process as a finite-horizon Markov Decision Process (MDP), following prior work (Black et al., 2023; Ren et al., 2024), where the diffusion planner serves as a stochastic policy that generates trajectories $\mathbf{x}$ conditioned on observations $o$ ; and the reward $r$ is directly given by $r = r_\phi(o, \mathbf{x})$.

To streamline the finetuning process, we adopt and modify GRPO method introduced by Guo et al. (2025) and calculate the estimated group relative advantage with group size $K$ as

$$\hat{A}_{\text{gr}}^{(k)} = \frac{r_k - \text{mean}\left(\{r_1, r_2, \cdots, r_K\}\right)}{\text{std}\left(\{r_1, r_2, \cdots, r_K\}\right)} \quad (8)$$

Since the reward signal is provided exclusively at the terminal trajectory, the RL loss is formulated as

$$\mathcal{L}_{RL}^\theta = -\mathbb{E}_{\mathcal{G} \sim \mathcal{P}_\theta}\left[ \frac{1}{KT} \sum_{k=1}^K \sum_{t=1}^T \log \pi_\theta\left(a_t^{(k)} \mid \psi_t^{(k)}\right) \gamma^{T-1-t} \hat{A}_{\text{gr}}^{(k)} \right] \quad , \quad (9)$$

where $\mathcal{G} = \{\tau^{(1)}, \ldots, \tau^{(K)} | \tau^{(k)} = (\psi_0^{(k)}, a_0^{(k)}, \ldots, \psi_{T-1}^{(k)}, a_{T-1}^{(k)}, \psi_T^{(k)})\}$ is a denoising trajectory group with the same observation $o$ and noise $z$. Here $\mathcal{P}_\theta$ is a mixed sampling distribution and $\gamma \in [0, 1]$ is a discount factor.

To curb overfitting on the small RL train set, we augment the RL objective with a behavior-cloning regularizer that keeps the updated policy close to a frozen pretrained reference. Specifically, following the approach outlined in Ren et al. (2024), the BC loss function is designed as

$$\mathcal{L}_{BC}^\theta = -\mathbb{E}_{\tau \sim \mathcal{P}_{ref}}\left[ \frac{1}{T} \sum_{t=0}^{T-1} \log \pi_\theta\left(a_t \mid \psi_t\right) \right], \quad (10)$$

where we sample state-action sequences $\tau$ using freezed initial model $\pi_{\text{ref}}$ to contruct mixed sampling distribution $\mathcal{P}_{ref}$ and encourage the finetuned model $\pi_\theta$ to maintain a high probability of producing the same actions as $\pi_{\text{ref}}$. The combined loss with weight coefficient $\alpha \geq 0$ is

$$\mathcal{L}^\theta = \mathcal{L}_{RL}^\theta + \alpha \mathcal{L}_{BC}^\theta. \quad (11)$$

After RL converges, we adopt a short supervised refresh that applies a few additional supervised learning epochs on dataset $\mathcal{D}_p$ to pull the policy back and erase residual drift. We employ the same loss function of Eq. 6. The DPGRPO algorithm is summarized in Algorithm 1, and more RL modeling details can be found in Appendix A.4.

# 5 EXPERIMENTS

## 5.1 EXPERIMENTAL SETUP

**Dataset and Metrics** We conduct experiments on three datasets: NavSim (Dauner et al., 2024), Internal Normal dataset for pretraining, and Internal Preference dataset for finetuning. On the NavSim benchmark, our generative trajectory model is evaluated following the official Predictive Driver Model Score (PDMS) (Dauner et al., 2024), which quantifies driving capacity by aggregating subscores for multiple objectives such as progress and comfort. On Internal Normal dataset, we pretrain diffusion policy with three distinct scales of training data, sampling frames at 2Hz to ensure temporal coverage. Internal Preference dataset, constructed in Appendix A.1, has two splits, which capture "aggressive" and "defensive" driving behaviors through human annotations, respectively. On Internal dataset, model performance is assessed using metrics including minADE, meanADE, minFDE and meanFDE.

**Human Preference Evaluation** We propose human preference judgments to evaluate trajectory output styles. Evaluators compare trajectory pairs under identical contexts and select the one that better aligns with the intended style. Based on this, we develop a systematic human evaluation framework and introduce the **Better or Equal Rate (BOE)**—the proportion of instances where one model is rated better than or equal to another (see Appendix B.2). This metric more precisely captures the evaluation objective centered on human preferences, which are inherently difficult to quantify.

The capacities of Internal dataset are shown in Appendix B.3 and more implementation details of our generative model, reward model and RL finetuning are presented in Appendix B.4.

## 5.2 MAIN RESULTS

### 5.2.1 NAVSIM BENCHMARK

Table 1: **Performance on *navtest* with closed-loop metrics.** "AR" represents auto-regressive training paradigm. "A/V" indicates the necessity of using anchors or vocabulary. Results using PDMS selector show the upper bound of TrajHF, marked in gray. All metrics are sourced from the cited paper.

| Method | Paradigm | A/V | NC ↑ | DAC ↑ | EP ↑ | TTC ↑ | C ↑ | PDMS ↑ |
|---|---|---|---|---|---|---|---|---|
| Transfuser (Chitta et al., 2022) | AR | × | 97.8 | 92.6 | 78.9 | 92.9 | 100 | 83.9 |
| Hydra-MDP (Li et al., 2024) | AR | ✓ | 98.3 | 96.0 | 78.7 | 94.6 | 100 | 86.5 |
| DiffusionDrive (Liao et al., 2024a) | Diffusion | ✓ | 98.2 | 96.2 | 82.2 | 94.7 | 100 | 88.1 |
| GoalFlow (Xing et al., 2025) | Diffusion | ✓ | 98.4 | 98.3 | 85.0 | 94.6 | 100 | 90.3 |
| TrajHF (Single sample) | Diffusion | × | 96.3 | 96.0 | 83.1 | 91.5 | 100 | 86.4 |
| **TrajHF (EM)** | Diffusion | × | 96.6 | 96.6 | 84.5 | 92.1 | 100 | 87.6 |
| TrajHF (PDMS selector) | Diffusion | × | 99.2 | 99.1 | 90.2 | 97.7 | 100 | 94.3 |

Table 1 reports the results of our model compared to other state-of-the-art models on the NavSim benchmark. We pretrain and finetune TrajHF on NavSim trainval split and utilize EM optimization from 15 candidate samples, detailed in Appendix B.4. Our approach demonstrates good performance across multiple metrics, including the comprehensive PDMS metric. Moreover, the architecture we propose does not rely on predefined anchors or vocabularies. This design covers a flexible and plausible trajectory planning distribution demonstrated by the results utilizing PDMS selector, which supports the downstream RLHF module. For fairness, we do not use Internal dataset for pretraining, and all methods adopt both camera and LiDAR as input.

### 5.2.2 HUMAN PREFERENCE FINETUNING

We evaluate RL finetuning method and supervised learning (SL) method on Internal normal and preference test splits. SL employs the same loss function of Eq. 6 as in the pretraining phase, while RL utilizes Algorithm 1. The comparative experimental results are summarized in Table 2.

Both SL and RL finetuning effectively adapt driving styles toward greater aggressiveness or defensiveness while reducing distance errors. When comparing methods, RL consistently outperforms SL with higher **BOE** score and lower minADE and minFDE values, demonstrating superior best-case

Table 2: **Performance of finetuning on Internal test splits with open-loop metrics.** "Diversity" is calculated as defined in Liao et al. (2024a). "BOE" is calculated over "w/o FT" model. "Aggr." and "Defen." refer to the aggressive and defensive datasets used for applying the respective methods. For "Normal" test split, baseline model w/o FT is not considered in comparison, marked in gray.

| Method | Test Split | minADE↓ | meanADE↓ | minFDE↓ | meanFDE↓ | Diversity↑ | BOE ↑ |
|--------|-----------|---------|----------|---------|----------|-----------|-------|
| w/o FT | Aggressive | 0.7546 | 1.5534 | 2.0054 | 4.3515 | 0.4419 | - |
| Aggr. SL | Aggressive | 0.5416 | **1.4485** | 1.3147 | **3.9245** | 0.5743 | 0.6577 |
| Aggr. RL | Aggressive | **0.5304** | 1.5036 | **1.2788** | 4.1039 | **0.6051** | **0.7660** |
| w/o FT | Defensive | 0.5538 | **1.4419** | 1.5854 | **4.3763** | 0.3741 | - |
| Defen. SL | Defensive | 0.3175 | 1.5317 | 0.8351 | 4.5239 | 0.3714 | 0.7430 |
| Defen. RL | Defensive | **0.3101** | 1.5309 | **0.8092** | 4.5603 | **0.3900** | **0.7922** |
| w/o FT | Normal | 0.2330 | 0.7734 | 0.5454 | 2.1207 | 0.3284 | - |
| Aggr. SL | Normal | 0.4612 | 1.3290 | 1.2850 | 3.8658 | 0.5276 | - |
| Aggr. RL | Normal | **0.3777** | **1.3034** | **1.0247** | **3.8057** | **0.5575** | - |
| w/o FT | Normal | 0.2330 | 0.7734 | 0.5454 | 2.1207 | 0.3284 | - |
| Defen. SL | Normal | 0.3675 | 3.0225 | 0.9428 | 8.7625 | **0.3811** | - |
| Defen. RL | Normal | **0.3361** | **2.5843** | **0.8537** | **7.4631** | 0.3773 | - |

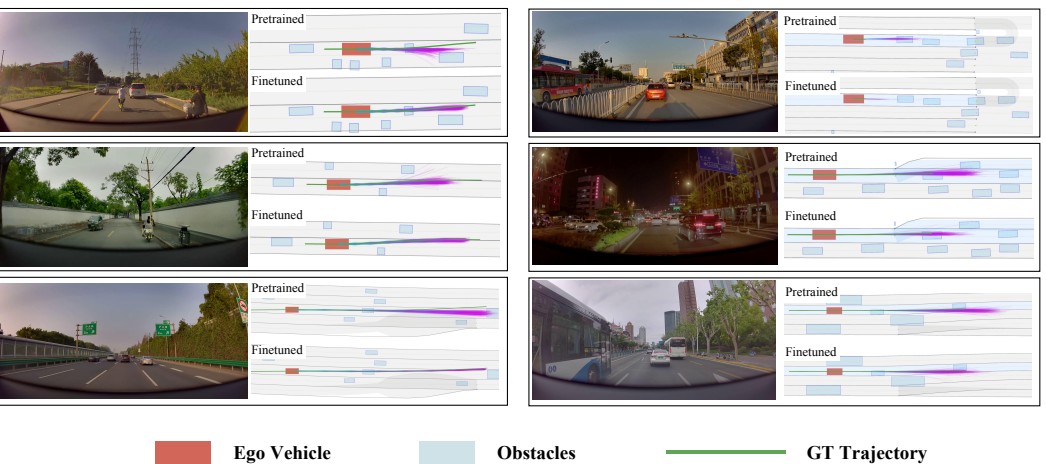

**Aggressive Finetuning**  **Defensive Finetuning**

🟥 **Ego Vehicle**   🟦 **Obstacles**   ── **GT Trajectory**

Figure 4: **Qualitative comparison of pretrained vs. RL finetuned diffusion policies.** On the right side of each case, the upper half and the lower half BEV images show the planned trajectories of the pretrained policy and the RL finetuned policy, respectively.

performance. **BOE** especially shows the effectiveness of TrajHF aligning with distinct personalized driving style. Additionally, RL generates trajectories with higher diversity metrics across preference datasets, producing more varied yet feasible driving behaviors. Furthermore, RL demonstrates greater practical utility by maintaining excellent performance on normal test splits despite preference-based finetuning, surpassing SL in nearly all metrics and preference indicators. This highlights RL's robust generalization capabilities across varied driving scenarios without significant performance degradation. Although SL achieves marginally better mean error metrics in some cases, this advantage likely stems from the inherent variance in RL method.

We conduct qualitative analysis to assess the effect of preference-aligned finetuning. As shown in Fig. 4, the finetuned model demonstrates improved context awareness and adaptability—proactively overtaking in complex scenarios for efficiency, or decelerating in ambiguous or risky situations for safety. These behaviors are absent in the pretrained policy, indicating that RL method effectively guides the model toward more realistic and human-aligned driving decisions. More detailed description and analysis for each visualized case are provided in Appendix B.5.

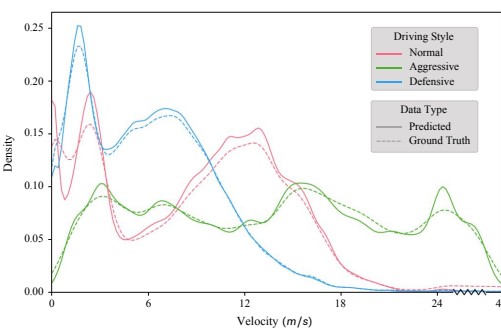

Figure 5: **Velocity distribution across driving styles.** Comparison of speed density profiles between model predictions (solid lines) and ground truth (dashed lines) for different styles.

Table 3: **Impact of Scaling the Training Set Size.** Here miA, meA, miF, meF and Div indicate minADE, meanADE, minFDE, meanFDE and Diversity, respectively.

| Scale | miA↓ | meA↓ | miF↓ | meF↓ | Div↑ |
|-------|------|------|------|------|------|
| 810K  | 0.40 | 1.09 | 0.87 | 2.67 | **0.28** |
| 3M    | 0.39 | 1.08 | 0.84 | 2.70 | 0.26 |
| 6M    | **0.37** | **1.04** | **0.81** | **2.66** | 0.26 |

Fig. 5 presents the velocity distributions for three driving styles from both model predictions and ground truth of test splits. The results show distinct velocity profiles across styles, demonstrating a positive correlation between stylization and vehicle speed. Specifically, the aggressive style shows higher probability density in high-speed regimes (24-40 m/s) than other styles. The finetuned model's predictions align well with the ground truth velocity distributions, demonstrating the effectiveness of our TrajHF framework to align the trajectory planning with personalized driving styles.

### 5.3 ABLATION STUDY

Table 3 illustrates the impact of training data scale in pretraining. The results are evaluated on the Internal normal test split. We observe that as the dataset size grows, the model exhibits non-marginal improvements in distance error metrics. However, this comes at the cost of reduced diversity, indicating that the diffusion policy learns a more concentrated distribution with larger data volumes.

In Appendix C, we report the ablation studies for Algorithm 1. These analyses include: (i) PPO-based variants derived from prior work and widely used approach (Appendix C.1); (ii) the effect of varying behavior cloning (BC) loss weights (Appendix C.2); and (iii) the influence of data scaling strategies (Appendix C.3). Collectively, these experiments substantiate the effectiveness and robustness of our proposed DPGRPO fine-tuning approach.

## 6 CONCLUSION

In this work, we introduce TrajHF, a human feedback-driven finetuning framework for generative trajectory models that aligns autonomous driving behaviors with diverse personal driving styles. By combining a DDPM-based multi-modal planner with reinforcement learning, TrajHF enables safe, feasible, and personalized trajectory generation. Experimental results demonstrate that TrajHF preserves key motion planning constraints while aligning to the specific human driving style. By leveraging human feedback as a supervisory signal, our approach addresses the limitations of traditional imitation learning, offering a more flexible and human-aligned trajectory generation paradigm.

**Limitations** The framework depends on human annotations, which can be subjective, inconsistent, and costly. Its generalization capacity is tied to the diversity and quality of collected preferences. Future work will explore auto-labeling via takeover data and scaling to broader contexts with richer social cues.

**Broader Impact** By generating more natural and socially aligned behaviors, our method can improve user trust and acceptability in autonomous vehicles. The framework may also extend to other domains requiring preference-aligned decisions. However, personalization in safety-critical systems introduces risks such as reinforcing unsafe or biased behaviors, requiring careful deployment and oversight.

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

# A DETAILS OF PREFERENCES ALIGNMENT METHOD

## A.1 PREFERENCE DATA COLLECTION

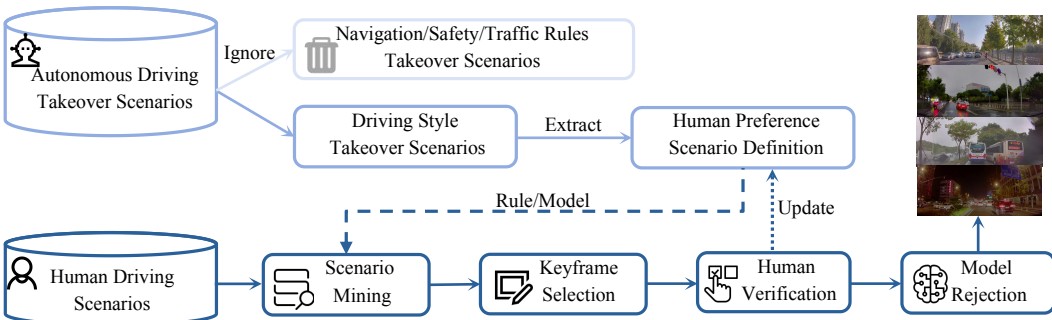

Figure 6: **The mining and annotation process for constructing a human driving preference scenario dataset.** Only takeover cases related to driving styles are used to extract human preference labeling, which is updated by data annotation results.

We propose an efficient process for data mining and automatic annotation to build a dataset $\mathcal{D}_p$ that reflects human driving preferences, as shown in Fig. 6. This process involves extracting human preference definition from takeover data, mining scenarios according to rules or models, selecting preference key frame and conducting manual selection and model rejection.

**Preference Scenario Mining** Human driving typically occurs in ordinary environments, making it difficult to define specific driving styles for every decision. For instance, choosing to overtake a slow-moving vehicle may not be classified as either aggressive or conservative. However, in critical scenarios—such as overtaking vehicles using the opposite lane, shown in Fig. 1—distinct driving behaviors can be identified. These critical situations, referred to as preference scenarios, are mined from large-scale human driver takeover data. The data is categorized into six classes (e.g., "too aggressive" or "too conservative"), each corresponding to a distinct driving style, which can be used to define rules or train models for identifying preference scenarios.

**Key Frame Selection** After identifying preference scenarios, only segments relevant to the preference need to be labeled. Rather than labeling every frame, we focus on key frames where significant actions occur, such as changes in speed or direction. If a frame is tagged too early, the defining action has not yet happened; if tagged too late, the action is already underway. Clear specifications for key frame identification allow for rule-based, automatic detection, enabling the potential large-scale annotation.

**Post-verification** The annotated key frames undergo random manual checks to ensure data quality. Human inspectors can update scenario definitions or introduce new preference scenarios in special cases. Model rejection uses baseline generative models to filter out the common cases that ground truth trajectories cover common average human preference knowledge.

## A.2 CONSTRUCTING SEMI-SYNTHETIC DATASET FOR REWARD MODEL TRAINING

Utilizing preference dataset $\mathcal{D}_p = \{(o^{(j)}; \mathbf{x}_c^{(j)})\}_{j=1}^n$ acquired by Appendix A.1, we adopt a semi-synthetic approach that enables generating multiple trajectory preference pairs within the same scenario $o$. We use the pretrained model to predict trajectories $\mathbf{x}_{raw}$ for each scenario, randomly selecting $q$ trajectories to form $q$ raw trajectory preference pairs $\{(\mathbf{x}_c^{(j)}, \mathbf{x}_{raw}^{(j)})\}_{j=1}^q$. Applied on whole dataset $\mathcal{D}_p$, we acquires raw preference-pair dataset $\mathcal{D}_{raw} = \{(o^{(j)}; \mathbf{x}_c^{(j)}, \mathbf{x}_{raw}^{(j)})\}_{j=1}^m$, where $m = q * n$.

However, trajectories directly generated by the model exhibit significant statistical differences from human-annotated trajectories, potentially leading to reward hacking during reward model training. To address this challenge, we present a trajectory reconstruction autoencoder $f_\eta$ that transfers the raw trajectory $\mathbf{x}_{raw}$ generated by our pre-trained model to a normalized trajectory $\mathbf{x}_i = f_\eta(\mathbf{x}_{\text{raw}})$. The

architecture of the model is a symmetric 1D convolutional autoencoder with two-layer encoder and decoder blocks using ReLU activations. The model utilizes a dual loss function:

$$\mathcal{L}^{\eta} = (1 - \delta)\,\mathcal{L}_{\text{recon}} + \delta\,\mathcal{L}_{\text{style}}, \tag{12}$$

$$\text{where } \mathcal{L}_{\text{recon}} = \|\mathbf{x}_i - \mathbf{x}_{raw}\|_2^2,$$

$$\mathcal{L}_{\text{style}} = \|\text{mean}(\mathbf{v}_i) - \text{mean}(\mathbf{v}_c)\|_2^2 + \|\text{std}(\mathbf{v}_i) - \text{std}(\mathbf{v}_c)\|_2^2.$$

The reconstruction loss $\mathcal{L}_{recon}$ maintains geometric consistency with the original diffusion-generated trajectories, and the style loss $\mathcal{L}_{style}$ transfers human-like motion characteristics by comparing velocity distribution statistics (means and variances) between generated and human-annotated trajectories.

We train the autoencoder $f_\eta$ on a small subset of $\mathcal{D}_{raw}$, apply $f_\eta$ to reconstruct $\mathbf{x}_{raw}$ and finally acquire semi-synthetic preference-pair dataset $\mathcal{D}_R = \{(o^{(j)}; \mathbf{x}_c^{(j)}, \mathbf{x}_i^{(j)})\}_{j=1}^m$.

## A.3 VALIDATION OF THE REWARD MODELS

In this section, we provide an explicit validation of the reward models using held-out test sets of human preference data. For each scene and driving style (defensive and aggressive), we construct a separate held-out set of human preference trajectory pairs.

At test time, the reward model assigns scores to both trajectories in each preference pair. A prediction is counted as correct if the model assigns the highest reward to the chosen trajectory. We report the number of correct predictions and the corresponding accuracy for each driving style in Table 4.

| Test Split | #Correct / #Total | Accuracy |
|---|---|---|
| Aggressive | 4028 / 4182 | 96.32% |
| Defensive | 3607 / 3615 | 99.78% |

Table 4: Validation results of the reward models on held-out human preference test sets.

These results demonstrate that the reward models are highly consistent with unseen human preferences, confirming their strong generalization performance.

## A.4 RL MODELING DETAILS

For a target driving style we collect a dataset $\mathcal{D}_p = \{(o, \mathbf{x}_c)\}$ distributed as $p(o, \mathbf{x}_c)$, where $o$ is an observation and $\mathbf{x}_c$ is a ground-truth trajectory. Given $o$, we draw a latent noise vector $z \sim \mathcal{N}(\mathbf{0}, \mathbf{I})$ and let the diffusion-policy $\pi_\theta$ generate a candidate terminal trajectory $\mathbf{x}_0 \sim \pi_\theta(\mathbf{x}_0 \mid o, z)$. Its scalar reward is then evaluated by $r_\phi(o, \mathbf{x}_0)$.

**Mixed sampling distribution** $\mathcal{P}_\theta$. Denote a length-$T$ denoising trajectory by $\tau = (\psi_0, a_0, \psi_1, a_1, \ldots, \psi_{T-1}, a_{T-1}, \psi_T)$, where the state–action pairs are defined as $\psi_t = (o, \mathbf{x}_{T-t})$ and $a_t = \mathbf{x}_{T-t-1}$. The joint distribution that produces all randomness in one rollout is

$$\mathcal{P}_\theta(\tau) = p(o)\,\mathcal{N}(z) \prod_{t=0}^{T-1} \pi_\theta(a_t \mid \psi_t)\,\delta[\psi_{t+1} = (o, a_t)], \tag{13}$$

where $\delta[\cdot]$ is the Dirac measure that enforces the deterministic state update, and $\mathbf{x}_0$ is obtained after the final denoising step $\psi_{T-1} \to a_{T-1}$.

**Optimization objective.** Using the mixed measure Eq. 13, the policy is trained to maximize

$$J(\theta) = \mathbb{E}_{\tau \sim \mathcal{P}_\theta}\big[r_\phi(o, \mathbf{x}_0)\big]. \tag{14}$$

**MDP expansion.** Following Black et al. (2023); Ren et al. (2024), the denoising process can be viewed as a $T$-step MDP

$$\begin{cases} \psi_t \triangleq (o, \mathbf{x}_{T-t}), \\ a_t \triangleq \mathbf{x}_{T-t-1}, \\ P(\psi_{t+1} \mid \psi_t, a_t) \triangleq \delta_{(o, a_t)}, \\ \pi_\theta(a_t \mid \psi_t) \triangleq p_\theta(\mathbf{x}_{T-t-1} \mid \mathbf{x}_{T-t}, o), \\ \rho_0(\psi_0) \triangleq p(o) \, \mathcal{N}(\mathbf{0}, \mathbf{I}), \\ R(\psi_t, a_t) \triangleq \begin{cases} r_\phi(o, \mathbf{x}_0), & t = T - 1, \\ 0, & \text{otherwise.} \end{cases} \end{cases} \tag{15}$$

Let $r_t(\psi_t, a_t) = \sum_{\tau=t}^{T-1} \gamma^{T-1-\tau} R(\psi_\tau, a_\tau)$ be the discounted return from step $t$. The REINFORCE loss is then

$$\mathcal{L}_{\mathrm{RL}}(\theta) = -\mathbb{E}_{\tau \sim \mathcal{P}_\theta}\Big[\sum_{t=0}^{T-1} \log \pi_\theta(a_t \mid \psi_t) \, r_t(\psi_t, a_t)\Big]. \tag{16}$$

**Variance reduction.** Instead of raw returns, we replace $r_t(\psi_t, a_t)$ in Eq. 16 with the advantage estimator

$$\hat{A}^{\pi_\theta}(\psi_t, a_t) = \gamma^{T-1-t}\Big(r_\phi(o, \mathbf{x}_0) - \hat{V}^{\pi_\theta}(\psi_{T-1})\Big), \tag{17}$$

where $\hat{V}^{\pi_\theta}$ is a learned critic. To avoid critic warm-up and extra hyperparameters, we subsequently adopt the GRPO method of Guo et al. (2025), which normalizes terminal rewards within each $K$-trajectory group and uses those estimated group-relative advantages $\hat{A}_{\mathrm{gr}}^{(k)}$ in place of $\hat{A}^{\pi_\theta}$, as shown in Eq. 8. And according to the reward definition and GRPO method, we simplified the REINFORCE loss Eq. 16 to Eq. 9.

# B DETAILS OF EXPERIMENTS

## B.1 EM ALGORITHM FOR TRAJECTORY OPTIMIZATION

Inspired by Varadarajan et al. (2021), we aim to find an optimal planning trajectory to represent the whole sampling trajectory distribution $\Psi$. This is achieved using an iterative clustering algorithm that functions similarly to Expectation-Maximization (EM) (Bishop & Nasrabadi, 2006), but with a hard assignment of cluster membership.

First, we randomly pick one trajectory generation sample as the cluster centroid, denoted as $\bar{\mu}$. The objective is to maximize the probability that a centroid sampled from $\Psi$ is located within a distance $d$ of at least one of the selected centroids. This selection criterion, which explicitly promotes trajectory distribution representation, is formulated as:

$$\bar{\mu} = \underset{\bar{\mu}}{\mathrm{argmax}} \sum_{k=1}^{K} q_k \max_{\bar{\mu}} \mathbb{I}(\|\mu_k - \bar{\mu}\|_2 \leq d) \tag{18}$$

where $q_k$ is the probability of sample $k$ from $K$ inferences and $\mathbb{I}(\cdot)$ is the indicator function.

Starting with the selected centroids, we iteratively update the parameters of $\bar{\bar{\Psi}}$ using an Expectation-Maximization-style algorithm. Each iteration consists of the following updates:

$$\bar{q} \leftarrow \sum_{k=1}^{K} q_k p(\mu_k; \bar{\Psi}) \tag{19}$$

$$\bar{\mu} \leftarrow \frac{1}{\bar{q}} \sum_{k=1}^{K} q_k p(\mu_k; \bar{\Psi}) x \tag{20}$$

$$\bar{\Sigma} \leftarrow \frac{1}{\bar{q}} \sum_{k=1}^{K} q_k p(\mu_k; \bar{\Psi})[\Sigma_k + (\mu_k - \bar{\mu})(\mu_k - \bar{\mu})^T] \tag{21}$$

Here, $p(x; \bar{\Psi})$ represents the probability that a sample $x$ is drawn from the model distribution specified by $\bar{\Psi}$.

## B.2 HUMAN PREFERENCE BOE

Evaluating driving style requires human-level semantic understanding of the current scene. For example, when there is a cyclist in front of the ego vehicle, considering the context, we can determine that overtaking the cyclist is a more aggressive behavior compared to following behind the cyclist. When using overtaking behavior as the primary criterion for aggressive style, we found that ADE and FDE metrics do not effectively reflect this behavior. Compared to an aggressive ground truth trajectory, if a model's trajectory is overall faster or slower longitudinally, or has lateral deviations, these would result in larger ADE and FDE values. However, these longitudinal and lateral deviations are not clearly associated with whether the model generates an overtaking trajectory. Therefore, we choose to introduce human preference judgments to evaluate trajectory output styles. Specifically, based on their subjective assessment, evaluators select which trajectory better meets style requirements from a pair of trajectories in the same context.

Based on this approach, we designed a systematic human evaluation framework to compare our models. To evaluate the performance of models $m_1$ and $m_2$ on a set of scenes $\mathcal{D}$, we conduct human evaluation experiments. For each scene $o \in \mathcal{D}$, the two models generate trajectories $\mathcal{T}_{m_1}(o)$ and $\mathcal{T}_{m_2}(o)$ respectively. Evaluators are asked to choose from three options without knowing the source of the trajectories (double-blind assessment): prefer the trajectory generated by $m_1$, prefer the trajectory generated by $m_2$, or consider both equally effective in meeting the specified driving style.

Formally, we define the human evaluation function $h(\mathcal{T}_{m_1}(o), \mathcal{T}_{m_2}(o))$, whose value is:

- 1 if the evaluator prefers $\mathcal{T}_{m_1}(o)$

- $-1$ if the evaluator prefers $\mathcal{T}_{m_2}(o)$

- 0 if the evaluator considers both equally effective (tie)

For each scene $s \in \mathcal{S}$, we collected evaluations from $N$ evaluators. Based on these evaluations, we define the metric **Better Or Equal Rate (BOE)**: The proportion where one model is considered better than or equal to the other model

$$\begin{cases} \text{BOE}_{m_1} = \dfrac{|\{o \in \mathcal{D} : h(\mathcal{T}_{m_1}(o), \mathcal{T}_{m_2}(o)) \geq 0\}|}{|\mathcal{D}|} \\ \text{BOE}_{m_2} = \dfrac{|\{o \in \mathcal{D} : h(\mathcal{T}_{m_1}(o), \mathcal{T}_{m_2}(o)) \leq 0\}|}{|\mathcal{D}|} \end{cases} \tag{22}$$

For the "aggressive" evaluation dataset, we manually select 53 clips from the test set where the pretrained model's performance exhibit significant deviations from the expected driving style. For the "defensive" evaluation dataset, we manually select 151 clips with similar characteristics. We recruit 5 professional annotators who regularly perform driving style labeling tasks to serve as evaluators. These evaluators are instructed to make preference judgments on trajectory outputs based on the specified driving style requirements and scene context. To mitigate individual bias, the preference

metrics for each model are derived by averaging the scores across all evaluators, ensuring a more objective assessment of the models' performance.

### B.3 INTERNAL DATA CAPACITIES

The table contains the capacity of each split in Internal dataset, depicted by scenario clips and frames.

Table 5: **Dataset capacities of different splits of Internal dataset.** Small, medium and large splits refer to Internal Normal dataset. "Aggressive" and "defensive" splits are from Internal Preference dataset.

| **Data Split** | Clips | Frames |
|---|---|---|
| small | 65,628 | 810K |
| medium | 298,706 | 3M |
| large | 487,812 | 6M |
| aggressive | 2967 | 35329 |
| defensive | 49030 | 136782 |

### B.4 IMPLEMENTATION DETAILS

Our model employs a base ViT (Dosovitskiy et al., 2020) as the image backbone and ResNet34 (He et al., 2016) as the LiDAR backbone. The input front-view images are concatenated at a resolution of $1024 \times 256$, while the LiDAR is $256 \times 256$. We adopt an SDE-based denoising paradigm with 10 steps for both training and inference. The learning rate is set to be $1e - 4$, and the batch size is 512, distributed across 8 NVIDIA-H20-RDMA GPUs. We use the $OneCycle$ scheduler and the Adam optimizer with default parameters. As required by the task, the model outputs 8 waypoints spanning 4 seconds. The model is trained for 1000 epochs as the base model on NavSim benchmark.

In the semi-synthetic dataset construction (Appendix A.2), the sampling rate $q = 3$. For the trajectory reconstruction model training, the weight of style $\delta = 0.3$, the learning rate is $1e - 3$, the batch size is 32 and the epoch number is 100. In reward model training, we adopt margin constant $m = 1$, the learning rate is $2e - 5$, the batch size is 32 and the max epoch number is 30 with early-stop mechanism.

For NavSim benchmark, we finetune the base model on NavSim trainval split using GRPO (Guo et al., 2025) and employ the PDMS metric as the reward. The pretrained model is fine-tuned for 45 epochs using 16 NVIDIA-H20-RDMA GPUs with a per-GPU batch size of 16, a BC loss weight of $1e - 2$, a learning rate of $6e - 5$ for the ViT component, and $5e - 5$ for the remaining parts. The EM algorithm iterates 25 times for each scenario.

For supervised learning (SL) and reinforcement learning (RL) experiment, we set sample number $K = 8$ and discount factor $\gamma = 0.99$. Behavior cloning loss weights $\alpha$ is set to $1e - 1$ in Section 5.2.2 and varies from 1 to $1e - 3$ in Appendix C.2. The learning rate is $5e - 5$ and the batch size is 128 for the SL and 32 for the RL. We adopt 200 epoch training for SL and 20 epoch training for RL. The short supervised refresh takes the same setting as SL and is trained for 100 epoch.

### B.5 QUALITATIVE ANALYSIS

In Fig. 4, TrajHF achieves excellent performance qualitatively for both "aggressive" and "defensive" styles.

For "aggressive" finetuning, in the first case, there are Vulnerable Road Users (VRU) and a parked vehicle ahead in the ego lane, and the ego vehicle is about to pass a vehicle in the oncoming left lane, representing a scenario with strong interactions. The pretrained policy plans trajectories that tend to follow behind the VRUs, showing a conservative behavior. However, after RL finetuning, the policy plans a trajectory that borrows the opposite lane to bypass the VRUs, becoming more aggressive

and efficient. The second case is similar to the first one, where the ego vehicle intends to bypass a VRU. Although the pretrained policy also plans lane-changing trajectories to bypass, their speeds are not fast enough. After RL finetuning, the policy significantly increases the driving speed. In the third case, the pretrained strategy plans trajectories following the slow-moving vehicle ahead in the current lane, exhibiting inefficient behavior. After RL finetuning, the planned trajectories involve a left lane change to overtake the preceding vehicle, thereby improving driving efficiency, making it more consistent with the aggressive human driving ground truth trajectory.

The effectiveness of finetuning with defensive driving preferences is evident in several qualitative cases. In the first and second scenarios, the finetuned model exhibits cautious behavior by proactively decelerating in potentially ambiguous or high-risk situations, such as approaching an intersection or navigating under low-visibility nighttime conditions. These adaptations reflect a heightened awareness of environmental complexity, which is not observed in the baseline model that maintains a higher velocity without accounting for contextual risk. In the third scenario, where a large vehicle initiates an overtaking maneuver, the defensive-style planner successfully generates a trajectory that yields space and reduces velocity to mitigate potential conflict. In contrast, the original model, lacking an explicit preference for conservative behavior, fails to adjust its trajectory and continues at its nominal speed, increasing the risk of lateral interaction. These results demonstrate that the preference-aligned model not only respects safety constraints but also anticipates socially appropriate responses to multi-agent interactions, contributing to more robust and human-aligned motion planning.

Additionally, we provide BEV video visualization in supplementary material. "videos.zip" contains three aggressive cases and three defensive cases same to Figure 6, named "aggressive_1.mp4" to "aggressive_3.mp4" from top to bottom and "defensive_1.mp4" to "defensive_3.mp4", respectively.

## B.6 ADDITIONAL EXPERIMENTS ON PREFERENCE DATASET

We show the results of collision rate and offroad rate to further monitor the safety performance after preference finetuning. The results of Aggressive test split are shown in Table 6.

Table 6: Collision rate and offroad rate on the aggressive test split before and after finetuning.

| Method | Collision Rate↓ | Offroad Rate↓ |
|---|---|---|
| w/o FT | 12.50% | 5.39% |
| Aggressive RL | **2.70%** | **1.59%** |

Contrary to the concern that aggressive driving might be unsafe, our results show that the collision rate on the "Aggressive" split drops significantly from 12.50% to 2.70%. This is because the "Aggressive" dataset consists of complex interaction scenarios (e.g., overtaking, cut-ins) mined from human takeover data. In these high-stakes situations, the Base Model often behaves hesitantly or fails to complete the maneuver in time, leading to failures. Also, offroad ricks are reduced after finetuning. By finetuning, TrajHF learns to be assertive and decisive—aligning with the expert human strategy—thereby navigating these interactions more safely and effectively. It is worth noting that the absolute collision rates and offroad rates reported above are influenced by inherent noise in the raw dataset, particularly inaccuracies in the ground truth yaw (heading) annotations, which makes precise collision calculation challenging even for ground truth trajectories. Therefore, we emphasize the relative trend over the absolute values.

We reimplement Transfuser (Chitta et al., 2022) on our preference dataset and train 100 epochs to measure the baseline performance, which is shown in Table 7.

Table 7: Performance of Transfuser (Chitta et al., 2022) on our preference dataset.

| Test split | minADE | min FDE | Collision Rate↓ | Offroad Rate↓ |
|---|---|---|---|---|
| Aggressive | 2.1377 | 5.4300 | 14.90% | 6.21% |
| Defensive | 1.1268 | 3.2822 | 8.66% | 0.20% |

TrajHF significantly surpasses Transfuser at all metrics, indicating the superior performance on the human preference alignment comparing to the baseline.

# C  SUPPLEMENTARY ABLATION EXPERIMENTS

## C.1  ABLATION FOR PREFERENCE FINETUNING ALGORITHMS

In this section, we present an ablation study on alternative preference finetuning algorithms for diffusion-based trajectory prediction. Preference finetuning in this setting remains a largely underexplored problem. The most closely related prior work is Wang et al. (2024). Since their reward model, dataset, and official implementation are not publicly available, we re-implement their PPO-based preference optimization procedure for comparison.

Since our task involves single-shot trajectory generation with only a terminal sequence-level reward signal, we adapt their algorithm to a critic-free variant formulation. We follow the description in Wang et al. (2024) and adapt it to our task by constructing a hybrid reward composed of an MSE-based trajectory consistency term and a preference score produced by a preference-trained reward model, which is also used in DPGRPO Algorithm 1. This hybrid reward is optimized using the standard clipped PPO objective together with a KL regularization term according to Wang et al. (2024). The results of both the preference test splits and the standard (normal) test splits are summarized in Table 8.

Table 8: Results of the preference finetuning variant adapted from Wang et al. (2024).

| Method | Test Split | minADE | meanADE | minFDE | meanFDE | Diversity |
|---|---|---|---|---|---|---|
| Aggr. Variant | Aggressive | 0.7099 | 1.5640 | 1.7958 | 4.2434 | 0.4704 |
| Aggr. Variant | Normal | 0.2716 | 0.8546 | 0.6518 | 2.3182 | 0.3397 |
| Defensive Variant | Defensive | 0.4638 | 1.3630 | 1.2667 | 4.1038 | 0.3754 |
| Defensive Variant | Normal | 0.3314 | 1.1452 | 0.8002 | 3.1791 | 0.3346 |

The results indicate that while the Wang et al. (2024) variant is able to perform preference fine-tuning to some extent, its performance on the preference test sets is consistently inferior to that of the DPGRPO-finetuned model. Meanwhile, its relatively stronger performance on the standard (normal) test sets suggests that this policy remains closer to the original pretraining distribution, indicating limited adaptation to target preference signals.

We conjecture that this Wang et al. (2024) variant method underperforms DPGRPO mainly for two reasons. First, the regularization and clipping strategy adopted from Wang et al. (2024), while effective for their simpler lane-change task, becomes overly restrictive in our complex multi-scenario setting, and the absence of an explicit critic may slightly limit stable credit assignment, jointly constraining the policy optimization. Such strong constraints risk collapsing the policy into a narrow mode, limiting its ability to capture the multimodal distributions that diffusion models can naturally generate and that diverse driving contexts require. In contrast, GRPO's group-advantage mechanism (Fig. 2) is better aligned with the multimodal nature of the diffusion generator, thereby more effectively unlocking its potential for preference tuning.

Furthermore, we evaluate Direct Preference Optimization (DPO) (Rafailov et al., 2023) as another alternative preference fine-tuning approach. The corresponding results are reported in Table 9.

Table 9: Results of the DPO-finetuned model.

| Method | Test Split | minADE | meanADE | minFDE | meanFDE | Diversity |
|---|---|---|---|---|---|---|
| Aggr. DPO | Aggressive | 4.9720 | 5.1282 | 10.2502 | 10.7709 | 0.1835 |

We find that DPO is ineffective for preference fine-tuning in autonomous driving. Due to the highly multi-objective nature of the task, DPO tends to overfit to a single dimension of the preference signal instead of balancing competing objectives, which often leads to training instability. This limitation is further amplified by DPO's sensitivity to noisy and ambiguous preference labels collected from diverse real-world takeover scenarios.

**Hyperparameter Settings for Wang et al. (2024) and DPO variants.**  Table 10 summarizes the key hyperparameters used for the Wang et al. (2024) variant and DPO in our experiments.

| Parameter | Wang et al. (2024) variant | DPO variant |
|---|---|---|
| Batch size | 32 | 32 |
| K sample | 8 | 8 |
| Learning rate | $5 \times 10^{-5}$ | $5 \times 10^{-5}$ |
| Max epoch | 30 | 30 |
| Epsilon Clipping | 0.2 | None |
| KL coef | 0.01 | None |
| Inverse temperature | None | 200 |

Table 10: Hyperparameter settings for Wang et al. (2024) and DPO variants

Regarding the Wang et al. (2024) variant setup, although we follow the overall algorithmic framework described in Wang et al. (2024), we do not directly reuse their original hyperparameter configuration. This is due to two major differences between the two settings. First, their work focuses on a lane-changing task, whereas our work addresses general trajectory prediction in diverse driving scenarios. These two tasks exhibit substantially different input distributions, behavioral variability, and optimization dynamics. Second, the model architectures are fundamentally different: Wang et al. (2024) employ an LSTM-based policy, while we use a diffusion-based trajectory generator. These architectures lead to different gradient scales, stability characteristics, and convergence behaviors, making direct hyperparameter transfer inappropriate.

To ensure a fair and controlled comparison, we therefore align the key hyperparameters with those used in our primary baseline DPGRPO to avoid confounding factors introduced by architectural and task mismatches, while maintaining meaningful comparability across different preference fine-tuning algorithms.

## C.2  ABLATION FOR BC LOSS WEIGHT

This table shows the ablation results for varying behavior cloning (BC) loss weights $\alpha$, which is tested on the Internal preference test split.

Table 11: Impact of Behavior Cloning Loss Weight.

| $\alpha$ | Data Split | minADE↓ | meanADE↓ | minFDE↓ | meanFDE↓ | Diversity↑ |
|---|---|---|---|---|---|---|
| 1.0 | Aggressive | 0.5535 | 1.5238 | 1.3422 | 4.1555 | 0.6035 |
| 0.5 | Aggressive | 0.5635 | 1.5747 | 1.3694 | 4.2782 | 0.6088 |
| 1e-1 | Aggressive | **0.5304** | **1.5036** | **1.2788** | **4.1039** | 0.6051 |
| 1e-2 | Aggressive | 0.5585 | 1.5697 | 1.3479 | 4.2692 | **0.6142** |
| 1e-3 | Aggressive | 0.5330 | 1.5332 | 1.2805 | 4.1650 | 0.6131 |
| 1.0 | Defensive | 0.3187 | 1.5219 | 0.8418 | 4.5173 | 0.3763 |
| 0.5 | Defensive | 0.3252 | **1.4671** | 0.8579 | **4.3304** | 0.3758 |
| 1e-1 | Defensive | **0.3101** | 1.5309 | **0.8092** | 4.5603 | **0.3900** |
| 1e-2 | Defensive | 0.3209 | 1.4827 | 0.8389 | 4.3797 | 0.3755 |
| 1e-3 | Defensive | 0.3195 | 1.5972 | 0.8368 | 4.6734 | 0.3782 |

The experimental results across multiple datasets demonstrate that $\alpha = 1e - 1$ consistently achieves optimal performance for both preference types. This optimal $\alpha$ balances exploration in preference space while preventing the policy from diverging too far from reasonable driving behaviors.

## C.3  ABLATION OF DATA SCALES OF DPGRPO

We add a data-efficiency ablation study. The results obtained with 10% and 50% random subsamples of the preference data are as follows:

Table 12: Impact of data scales of DPGRPO. "Aggr." and "Defen." refer to the aggressive and defensive datasets used for training and evaluation.

| Method | Scale | Test Split | minADE↓ | meanADE↓ | minFDE↓ | meanFDE↓ | Diversity↑ |
|--------|-------|-----------|---------|----------|---------|----------|-----------|
| Aggr. RL | 10% | Aggr. | 0.7025 | 2.1477 | 1.6146 | 5.5560 | **0.6437** |
| Aggr. RL | 50% | Aggr. | 0.5431 | 1.5623 | 1.3091 | 4.2650 | 0.6015 |
| Aggr. RL | 100% | Aggr. | **0.5304** | **1.5036** | **1.2788** | **4.1039** | 0.6051 |
| Defen. RL | 10% | Defen. | 0.4483 | 1.8836 | 1.1182 | 5.2600 | 0.3746 |
| Defen. RL | 50% | Defen. | 0.3352 | 1.5785 | 0.8638 | 4.6677 | **0.4014** |
| Defen. RL | 100% | Defen. | **0.3101** | **1.5309** | **0.8092** | **4.5603** | 0.3900 |

The experiments reveal a logarithmic saturation trend: performance improves substantially when the data budget increases from 10% to 50%, whereas the marginal gain from 50% to 100% is much smaller. This aligns with the normal data-efficiency behavior in reinforcement learning and underlines the effectiveness of the proposed DPGRPO algorithm.

### C.4 ABLATION OF SHORT SFT REFRESH

In this section, we isolate the contribution of the short SFT)refresh phase in DPGRPO. Specifically, we compare the full DPGRPO framework with a GRPO-only variant, in which the short supervised refresh phase is completely removed, and the policy is trained solely using GRPO-style reinforcement learning updates.

The quantitative comparison results are summarized in Table 13. Compared to the full DPGRPO model, the GRPO-only variant exhibits consistently worse preference-alignment performance across both aggressive and defensive test splits. In addition, the trajectory diversity is also reduced.

Table 13: Ablation Results of GRPO-only Method

| Method | Test Split | minADE | meanADE | minFDE | meanFDE | Diversity |
|--------|-----------|--------|---------|--------|---------|-----------|
| Aggr. GRPO | Aggressive | 2.79 | 6.87 | 8.08 | 19.14 | 0.67 |
| Aggr. GRPO | Normal | 1.00 | 3.32 | 3.11 | 9.90 | 0.64 |
| Defensive GRPO | Defensive | 0.84 | 3.38 | 1.78 | 8.05 | 0.51 |
| Defensive GRPO | Normal | 0.63 | 4.29 | 1.07 | 9.93 | 0.49 |

These results confirm that the short supervised refresh phase is a critical component of DPGRPO, which effectively restores general-purpose driving behaviors, recovers performance in standard test splits, and simultaneously maintains strong alignment with the target driving style.

## D   LLM USAGE

We utilize Large Language Models during the writing process. Their application is strictly confined to polishing the language and syntax at the discretion, with the goal of optimizing expression without influencing the intellectual content.

