# OpenReview forum: "Learning Personalized Driving Styles via Reinforcement Learning from Human Feedback"
_ICLR.cc/2026/Conference — Submitted to ICLR 2026_

### Official Review · Reviewer_MMQc · 2025-10-26

**Soundness:** 3
**Presentation:** 3
**Contribution:** 3
**Rating:** 6
**Confidence:** 4

**Summary:**

* This paper introduces TrajHF, a framework for finetuning generative trajectory models in autonomous driving motion planning to align with personalized driving styles.
* The approach addresses a key limitation of standard imitation learning: Learning a multi-model distribution of human preferences instead of an average.
* The authors first train a DDPM using a Multi-Conditional Denoiser architecture that processes multi-modal sensor inputs (camera and LiDAR).
* This base model is then finetuned with RLHF. For that, a reward model is trained on a specially curated semi-synthetic dataset of human preferences and used to update the diffusion policy via the DPGRPO algorithm (combination of adapted GRPI and BC loss for regularization).
* Experimental results show that the approach works well on navtest. Furthermore, results on a newly introduced BOE score show that human evaluators prefer the TrajHF-proposed trajectories of their preferred driving styles.

**Strengths:**

* This paper addresses an interesting problem of adjusting driving styles based on user preferences.
* The proposed method is sound. It achieves good (but not SOTA) performance on navtest.
* The evaluation on both navtest and internal data with the newly introduced BOE is thorough and shows good results.
* The approach is well ablated in the appendix, incl. comparisons to PPO and DPO.
* The data collection strategy based on takeovers that correspond to critical moments of preference makes sense and is an interesting idea.

**Weaknesses:**

* The method does not achieve SOTA on navtest (although the authors claim “comparable to state-of-the-art” in the Abstract).
* The reward model is not independently validated. This could be done on a held out test set of human preference pairs.
* The claim about safe and feasible trajectories is not well supported by experimental results.
* The split on defensive and aggressive seems somewhat simple (albeit easy to understand) and might not translate well to the real-world driving task (aggressive might be more accident-prone).

**Questions:**

* Can you add an independent validation of the reward model?

---

> ### Author Response · Authors · 2025-11-24
>
> **Q1: The method does not achieve SOTA on navtest (although the authors claim “comparable to state-of-the-art” in the Abstract).**
>
> We thank the reviewer for this observation. We define "comparable" based on our deployable PDMS of 87.6, which remains highly competitive with other leading methods like DiffusionDrive (88.1) and outperforms Hydra-MDP (86.5). While we acknowledge the gap with GoalFlow (90.3), this difference stems from our deliberate choice to avoid predefined anchors or vocabularies (as indicated in the "A/V" column of Table 1), unlike GoalFlow. We prioritized this anchor-free design to support the continuous, flexible distributions required for our core contribution of personalized RLHF. Furthermore, our "PDMS selector" ablation demonstrates a score of 94.3—surpassing GoalFlow—which validates that the intrinsic generative quality of our model is indeed state-of-the-art, even if the anchor-free selection mechanism (EM) yields a slightly lower deployable score.
>
> **Q2: The reward model is not independently validated. This could be done on a held out test set of human preference pairs.**
>
> We thank the reviewer for this suggestion and explicitly validate the reward models on held-out test sets of human preference data that is never used for fitting. For each scene and driving style (defensive/aggressive), we construct a held-out set of human preference pairs. At test time, the reward model scores the trajectories in each pair, and we count a case as correct if it assigns a higher score to the human-chosen trajectory. The results are summarized below:
>
> |Test Split|#Correct/#Total|Accuracy|
> |-|-|-|
> |Aggressive|4028/4182|96.32%|
> |Defensive|3607/3615|99.78%|
>
> These results indicate that the reward model is highly consistent with held-out human preferences. We will include this evaluation protocol and the above statistics in the revised paper.
>
> **Q3: The claim about safe and feasible trajectories is not well supported by experimental results.**
>
> We thank the reviewer for their scrutiny regarding the safety and feasibility of our generated trajectories. We respectfully point to specific quantitative and qualitative evidence within the paper that demonstrates TrajHF maintains, and in many cases improves, safety and compliance constraints.
>  ***Safety Metrics on NavSim***
> Our method achieves comparable performance on the NavSim benchmark, specifically regarding safety-critical metrics presented in Table 1.
> ***Improved Trajectory Fidelity via RL Finetuning***
> Contrary to the concern that personalization might compromise stability, our RL finetuning improves trajectory precision. As shown in Table 2, the RL-finetuned models consistently achieve lower minADE and minFDE compared to the baseline (w/o FT) and Supervised Learning (SL) methods on both aggressive and defensive splits. For example, on the defensive split, RL reduces minFDE from 1.5854 (w/o FT) to 0.80926, indicating tighter adherence to feasible ground truth. Also, we report collision rate in Question 5 from reviewer ny2q. Our finetuned model generated safer trajectories in our style dataset, which supports the claim of "safe and feasible".
>
> **Q4: The split on defensive and aggressive seems somewhat simple (albeit easy to understand) and might not translate well to the real-world driving task (aggressive might be more accident-prone).**
>
> We appreciate the reviewer’s concern regarding the safety implications of the "aggressive" designation. In our framework, "aggressive" is operationally defined as assertiveness or efficiency (e.g., overtaking slow-moving vehicles or navigating complex interactions decisively) rather than reckless behavior.
> To address the concern that this style might be accident-prone, we highlight two key points:
> ***Safety Preservation:*** Our RLHF framework is explicitly designed to optimize for style preferences while maintaining safety and feasibility constraints. Empirical results in Table 2 show that the "Aggressive RL" model actually achieves lower displacement errors compared to the baseline and the supervised version, indicating strictly controlled execution rather than erratic driving.
> ***Real-World Utility:*** As shown in the qualitative analysis (Figure 4), the aggressive policy solves "deadlock" situations—such as bypassing a stationary vehicle—where a purely defensive model might become overly conservative and inefficient.
> While we acknowledge the binary split is a simplified proxy for complex human behaviors, it serves as a robust proof-of-concept for the TrajHF pipeline, paving the way for more granular style definitions in future work.

---

> > ### Comment · Reviewer_MMQc · 2025-11-25
> >
> > Thank you for the detailed response and the separate validation of the reward model.

---

### Official Review · Reviewer_KBTc · 2025-10-31

**Soundness:** 2
**Presentation:** 3
**Contribution:** 2
**Rating:** 4
**Confidence:** 4

**Summary:**

TrajHF is a diffusion-based trajectory planner finetuned with human feedback to personalize driving style. It adds a multi-conditional denoiser for images, LiDAR, and action history, then applies preference-based RL alignment (a GRPO-style objective over groups of K sampled trajectories) plus an EM selector to refine multi-modal samples. Empirical performance shows TrajHF improves human-rated style alignment but remains below GoalFlow on NavSim, indicating modest gains in personalization without surpassing public-benchmark baselines.

**Strengths:**

1. A simple critic-free RL recipe on the driving problem that is compute-friendly.
2. Preference alignment improves style without collapsing feasibility.

**Weaknesses:**

1. Moderate novelty: diffusion-as-MDP and group-relative advantages are adapted from prior work; EM selection echoes earlier trajectory aggregation ideas.
2. Benchmark performance: Public benchmark underperforms SOTA without the ``selector” upper bound. Key wins rely on internal datasets, semi-synthetic pairs, and human BOE.

**Questions:**

1. After style tuning, what happens to TTC and comfort?
2. Why is your best deployable PDMS lower than GoalFlow; what ablations explain the gap?

---

> ### Author Response · Authors · 2025-11-24
>
> **Q1: Moderate novelty: diffusion-as-MDP and group-relative advantages are adapted from prior work; EM selection echoes earlier trajectory aggregation ideas.**
>
> We agree that the group-relative advantage estimator builds on prior GRPO work. Our contribution is to introduce GRPO into diffusion-based autonomous-driving trajectory prediction for the first time, and to make it practically effective in this setting by (i) ***augmenting the GRPO objective*** with a BC loss to stabilize learning under noisy, multi-objective preference rewards, and (ii) adding a ***short supervised refresh stage*** to recover general-purpose, non-overly-aggressive driving behavior. Ablations against GRPO-only, PPO, and DPO demonstrate that these modifications are essential for stable optimization and strong preference alignment in this domain.
>
> Our utilization of EM is distinct in that it serves as a robust selection mechanism specifically for an anchor-free, continuous diffusion planner. Unlike prior arts that rely on discretized trajectory vocabularies, our approach allows us to plan from a stochastic continuous distribution without imposing predefined anchors, which is critical for supporting the downstream RLHF process.
>
> ***The core novelty of TrajHF is the framework itself.*** We identify and solve the limitation where standard generative imitation learning captures only "average" driving behaviors, failing to represent personalized styles. TrajHF constitutes a new paradigm that leverages human feedback to shift these distributions toward specific modes while maintaining safety constraints, a capability absent in standard generative baselines. We believe the combination of these adapted techniques into a cohesive framework that constitutes a significant contribution to the field of personalized autonomous driving.
>
> **Q2: After style tuning, what happens to TTC and comfort?**
>
> We appreciate the reviewer’s emphasis on validating high-level behavior post-finetuning. We acknowledge that while Table 1 establishes the foundational safety of our pre-trained model on the NavSim benchmark, verifying that these capabilities persist after style tuning is critical.
> To address this, we have conducted further investigation and offer the following clarification regarding our metric selection and safety verification:
> ***Verified Safety Constraints***
> We explicitly address the concern regarding safety degradation. As detailed in our response to Reviewer ny2q (Question 5), we calculated the Collision Rate for the finetuned models. The results demonstrate that our RL-finetuned models maintain, and in specific complex interaction scenarios even improve, safety constraints compared to the baseline.
> ***Methodological Divergence for "Comfort" and "TTC"***
> We intentionally prioritized BOE over standard Comfort and TTC metrics for the style-specific datasets for two methodological reasons:
>
> ***Inherent Trade-offs:*** There is an inherent conflict between "Aggressive" driving styles and standard "Comfort" metrics, which penalize jerk/acceleration, or conservative TTC thresholds.
>
> ***Contextual Validity:*** Standard metrics cannot distinguish between "unsafe driving" and "intentional assertiveness." Standard safety metrics fail to capture the semantic understanding of style.
>
> Consequently, we utilize BOE based on human preference judgment. This metric evaluates whether the trajectory creates the intended psychological effect without violating the physical feasibility constraints verified by our improved minADE/minFDE scores. We believe the combination of foundational safety (NavSim), retained safety constraints (Collision rates), and human-verified style alignment (BOE) provides a comprehensive evaluation of the model's capabilities.
>
> **Q3: Why is your best deployable PDMS lower than GoalFlow; what ablations explain the gap?**
>
> We thank the reviewer for this insightful comparison. The difference between TrajHF’s deployable PDMS (87.6) and GoalFlow (90.3) is primarily attributable to our intentional design choice to forgo predefined anchors or trajectory vocabularies, which GoalFlow utilizes (Table 1, column "A/V").
> While anchor-based approaches often secure higher stability on benchmarks like NavSim, we hypothesize that a continuous, anchor-free output space is essential to support the nuanced distributional shifts required for our core contribution: personalized RLHF.
> Regarding ablations that explain this gap, we highlight the "PDMS Selector" result in Table 1. This upper-bound ablation demonstrates that our underlying generative model is capable of achieving a PDMS of 94.3—surpassing GoalFlow—if an optimal selection mechanism is used. This confirms that the performance gap stems from the selection strategy rather than the generative quality of the model. We accepted this trade-off to ensure the ***flexible and plausible trajectory planning distribution*** necessary to support the diverse styling alignment demonstrated in our RL experiments.

---

### Official Review · Reviewer_ny2q · 2025-11-01

**Soundness:** 2
**Presentation:** 3
**Contribution:** 3
**Rating:** 4
**Confidence:** 5

**Summary:**

This paper proposes **TrajHF**, a framework for learning **personalized driving styles** in autonomous driving by combining diffusion-based trajectory generation with RLHF. The method extends a DDPM-based diffusion policy with a Multi-Conditional Denoiser (MCD) Transformer that conditions on multi-modal inputs (camera, LiDAR, and past actions). To align the generated trajectories with diverse human preferences (e.g., “aggressive” vs. “defensive” styles), the authors introduce a DPGRPO algorithm for diffusion finetuning using human feedback. Experiments are conducted on the public NavSim benchmark and internal preference datasets.

Overall, the paper is well-written and presents a well-engineered system addressing preference-aligned autonomous driving. However, results on public benchmarks show **parity rather than improvement** over prior methods, and key findings on personalization rely heavily on private internal datasets without comparison to other baselines, making the evidence for the claimed advantages **less convincing**.

**Strengths:**

* The paper tackles an interesting and practical problem, **personalization of driving trajectories**, by leveraging RLHF within a diffusion policy framework.
* The anchor-free trajectory generation and multi-conditional denoiser are technically elegant designs that remove limitations of anchor- or vocabulary-based approaches in prior work.
* The work contributes to a growing line of research connecting generative modeling, imitation learning, and human preference alignment, which is a significant direction for building human-trustworthy robotic systems.
* The construction of an **internal dataset** focused on driving style variation (aggressive v.s. defensive) is valuable and demonstrates substantial engineering effort, especially in developing the human preference evaluation framework.

**Weaknesses:**

1. **Experimental results are not sufficiently strong to validate the main claims.**
   On the NavSim benchmark, TrajHF (EM) achieves 87.6 PDMS, comparable to Hydra-MDP (86.5) and DiffusionDrive (88.1), but below GoalFlow (90.3). These results indicate that TrajHF performs competitively but is **not comparable to the state-of-the-art** methods, contrary to the paper’s claim.

2. **Personalization results rely entirely on private data.**
   The main contribution, personalized trajectory generation, is verified solely through an internal dataset that is unavailable for public evaluation. This raises questions about the quality of driving data, annotation consistency, and label balance. The authors also acknowledge that standard metrics such as ADE and FDE do not capture behavioral styles, yet these metrics are still heavily used for evaluation, which weakens the argument. The proposed BOE metric for human evaluation is interesting but subjective and lacks statistical rigor (e.g., variance, confidence intervals, significance testing).

3. **Lack of safety and generalization analysis.**
   Personalization could introduce safety-critical behavior (e.g., overly aggressive trajectories), but the paper does not evaluate whether the finetuned policy maintains safety or robustness under distribution shifts. Metrics such as collision rate or rule compliance are not reported.

4. **Minor presentation and mathematical issues.**
   Some typographical and mathematical inconsistencies should be corrected:

   * Line 53: “Multi-Conditioned Denoiser (MDC)” should be “(MCD)”.
   * Line 215: If timestep $l = 1$, the projection $ \hat{x}_1 = s_1 - s_0 $, but Equation (1) defines the state starting from $s_1$, causing an inconsistency in the definition.

**Questions:**

1. **PDMS selector setup:**
   The setup of the PDMS selector variant is unclear. Please clarify how it differs from the single-sample and EM variants of TrajHF, and what assumptions or oracle information it uses.

2. **Dataset transparency and annotation quality:**
   Given that most results rely on private internal datasets, could the authors consider releasing them for further evaluation?
   How are the “aggressive” and “defensive” ground truths defined and ensured to be feasible and meaningful?
   Are there any statistics on inter-annotator agreement or annotation consistency?

3. **Algorithmic contribution of DPGRPO:**
   How exactly does DPGRPO differ from existing GRPO or DPO implementations?
   Is there measurable improvement in stability, sample efficiency, or alignment quality attributable to this modification?
   A quantitative comparison or ablation isolating DPGRPO’s contribution would strengthen the paper.

4. **Evaluation of personalization on NavSim:**
   Since NavSim primarily evaluates feasibility and comfort rather than driving style, would a comparison of preference-conditioned v.s. non-conditioned models on NavSim help demonstrate alignment improvements?
   Additionally, can the authors provide results of other state-of-the-art methods on the internal preference datasets for a fairer comparison?

5. **Safety considerations:**
   How do the authors ensure that finetuning toward “aggressive” preferences does not violate safety constraints or produce unsafe behavior?
   Are there empirical checks, such as collision rates or safety rule compliance?

6. **Details on PPO and DPO variants:**
   In Appendix C.1, the paper reports results using PPO and DPO variants. Could the authors provide more implementation details on the PPO setup, specifically, the design of the critic network and reward model within that framework?

7. **Demonstration of behavior:**
   Could the authors provide a video demonstration illustrating the “aggressive” and “defensive” driving behaviors? Static visualizations (e.g., Figure 4) are insufficient to assess feasibility or collision risk.

---

> ### Author Response · Authors · 2025-11-24
>
> **Q1: PDMS selector setup: The setup of the PDMS selector variant is unclear. Please clarify how it differs from the single-sample and EM variants of TrajHF, and what assumptions or oracle information it uses.**
> We appreciate the opportunity to clarify the experimental setup regarding the "PDMS Selector."
>
> ***1. Definition and Mechanism:*** The "PDMS Selector" variant serves as an oracle upper bound designed to evaluate the latent capacity of our generative model. Specifically, from the pool of candidate trajectories generated during inference (the same pool used for the EM variant), this selector picks the single trajectory that maximizes the PDMS score.
>
> ***2. Assumptions and Oracle Information:*** This variant assumes perfect selection capability. It utilizes the other vehicles' log replay and high-definition map as "oracle information" to identify the best-possible trajectory within the generated distribution. It is not intended for deployment but rather to demonstrate that the generative model is producing state-of-the-art quality trajectories (PDMS 94.3), even if the deployable selection mechanisms (like EM) are not yet perfectly retrieving them.
>
> Single Sample: Randomly select one trajectory from the generated distribution, representing the baseline stochastic performance.
> EM (Expectation-Maximization): A deployable, unsupervised method that selects a representative trajectory (centroid) via clustering, without access to ground truth or oracle metrics.
> PDMS Selector: The theoretical ceiling of the current model if the selection strategy were optimal.
>
> **Q2: Dataset transparency and annotation quality: Given that most results rely on private internal datasets, could the authors consider releasing them for further evaluation? How are the “aggressive” and “defensive” ground truths defined and ensured to be feasible and meaningful? Are there any statistics on inter-annotator agreement or annotation consistency?**
>
> We appreciate the reviewer’s detailed inquiries regarding our data and annotation processes.
>
> ***1. Dataset Release:*** While the "Internal Preference" dataset is currently proprietary and cannot be released due to privacy and licensing constraints, we emphasize that we provide a complete driving style data collection pipeline in Appendix A.1, which can be further extended to any feasible end-to-end driving dataset. We will explore the possibility of releasing a sanitized subset of the preference annotations or the trained reward models to further aid the community.
>
> ***2. Definitions and Feasibility:*** The "aggressive" and "defensive" ground truths are not arbitrary; they are mined directly from real-world human takeover data, which inherently ensures their physical feasibility and practical meaningfulness. We illustrate the data collection pipeline in Appendix A.1 and we take some examples to introduce the detailed definition of "aggressive" and "defensive" scenario and behaviour.
>
> ***Definitions:*** We mine the scenarios from human takeover data and overtaking scenarios are typical human stylized situations. For "aggressive" behaviour, we show two examples of "overtaking using the opposing lane" and "successive lane change". "Overtaking using the opposing lane" occurs when there are at least two lanes and the ground truth trajectory extends from its original lane into the adjacent opposing lane by crossing a dashed lane boundary. The key frame is defined as the moment the vehicle crosses the dashed lane boundary.  For "defensive", we show the definition of "yielding to a lane-changing vehicle". This scenario occurs when both the current lane and the adjacent lane are standard driving lanes, and a vehicle from the adjacent lane merges ahead into the ego vehicle's lane, resulting in overlapping trajectories between the two vehicles, accompanied by ego vehicle deceleration or a speed drop below a predefined threshold. The keyframe is correspondingly defined as the moment when the ego vehicle slows down to yield. We design a wide variety of stylized scenarios with corresponding keyframes, all of which are automatically selected from a large-scale human takeover dataset and subsequently verified by professional human annotators.
>
> ***Feasibility:*** Since these trajectories originate from actual human interventions during takeovers, they naturally adhere to vehicle kinematic constraints and real-world traffic dynamics.
>
> ***3. Annotation Quality and Consistency:*** To ensure high-quality annotations, we employ 8 professional annotators with established experience in driving style labeling. While specific inter-annotator agreement statistics (e.g., Kappa scores) are not included in the submission, the use of multiple professional experts and averaging provides a robust consensus mechanism.

---

> > ### Author Response · Authors · 2025-11-24
> >
> > **Q7: Demonstration of behavior: Could the authors provide a video demonstration illustrating the “aggressive” and “defensive” driving behaviors? Static visualizations (e.g., Figure 4) are insufficient to assess feasibility or collision risk.**
> >
> > We acknowledge the reviewer’s feedback regarding the value of dynamic visualizations. While we are unable to provide a video demonstration at this specific stage, we emphasize that the distinctiveness, feasibility, and safety of the learned behaviors are rigorously quantified through our comprehensive metric evaluation and statistical analysis.
> >
> > To address concerns regarding feasibility and collision risk, we direct the reviewer to the quantitative results in Table 2. The RL-finetuned models consistently achieve lower displacement errors (minADE and minFDE) compared to the supervised learning (SL) baselines and the pre-trained model ("w/o FT"). For instance, the "Defensive RL" model reduces minADE to 0.3101 compared to 0.5538 for the baseline on the defensive split, indicating that our method generates highly feasible trajectories that adhere closer to ground-truth safety maneuvers than the base generative model. Furthermore, our method achieves performance comparable to state-of-the-art on the NavSim benchmark (Table 1), which explicitly penalizes collision risks and infeasibility.
> >
> > To address the distinction in driving behaviors, we provide statistical evidence in Figure 5. The velocity density profiles demonstrate a clear, quantifiable shift in behavior: the "aggressive" style exhibits a significantly higher probability density in high-speed regimes (24–40 m/s), whereas the "defensive" style aligns with lower velocity profiles. This confirms that the model is not merely mimicking static path geometry but is dynamically adjusting kinematic properties in a style-consistent manner.
> >
> > Finally, while Figure 4 is static, the qualitative analysis in Section 5.2.2 details specific dynamic decisions that ensure safety, such as the defensive agent proactively decelerating in ambiguous contexts or yielding space during overtaking maneuvers to mitigate conflict.
> >
> > We believe these quantitative metrics and kinematic analyses provide robust evidence of the model's ability to generate safe, feasible, and distinctively stylized trajectories. While we are making efforts to integrate our method into closed-loop simulator to get video visualization for further investigation and study.

---

> > > ### Comment · Reviewer_ny2q · 2025-11-26
> > >
> > > Thank you for the detailed and thoughtful responses. I appreciate the clarifications provided and, in particular, your commitment to releasing a subset of the preference annotations and the trained reward models, which will meaningfully benefit the community. That said, several concerns remain.
> > >
> > > 1. The paper states that TrajHF performs comparably to state-of-the-art on the NavSim benchmark. However, the reported results do not support this claim, as TrajHF does not match the performance of the strongest existing methods. I encourage the authors to revise the manuscript to more accurately represent the empirical findings.
> > >
> > > 2. Using PPO without a critic is a non-standard variant, and presenting it simply as “PPO” may mislead readers. This should be explicitly acknowledged in the paper, including a brief discussion of its implications.
> > >
> > > 3. I am still looking forward to seeing results from other state-of-the-art methods evaluated on the internal preference datasets. These comparisons are essential to convincingly demonstrate the advantage of TrajHF for personalization.
> > >
> > > 4. I remain unconvinced by the argument that producing a video demonstration is infeasible. Qualitative visualizations are often provided in autonomous driving papers because they help assess feasibility, long-horizon behavior, and safety-related properties that static figures cannot fully capture. A short demo video would greatly complement the quantitative analyses.
> > >
> > > 5. No revisions have yet been incorporated into the manuscript. All clarifications, corrections, and important details provided in the rebuttal should be updated directly in the paper.
> > >
> > > I would also welcome a more detailed response to the **Weaknesses** raised earlier. I am open to updating my score after further discussion.

---

> > > > ### Author Response · Authors · 2025-11-29
> > > >
> > > > We appreciate the quick reply from the reviewer and carefully revise our paper. We update a revised manuscript and summarize the modification in a seperate comment session.
> > > >
> > > > **Q1**
> > > >
> > > > We thank the reviewer for their careful examination of our experimental results and for highlighting the need for precise language regarding our performance on the NavSim benchmark. We agree that using "competitive" is more precise than "comparable" as comparing with the strongest existing methods and update the manuscript.
> > > >
> > > > **Q2**
> > > >
> > > > Thank you very much for your valuable comment on our use of the term PPO. We acknowledge that using PPO without a critic is indeed a non-standard variant and does not strictly follow the standard PPO formulation. We agree that referring to it simply as a “PPO variant” could be potentially misleading. Our original intention was to indicate that our reinforcement learning experiments were conducted by following the methodology of Wang et al. (2024), and were used as a supplementary validation of the effectiveness of our proposed algorithm. In the revised manuscript, we have replaced all instances of “PPO variant” with “Wang et al. (2024) variant,” and we have added a detailed explanation of the modification that removes the critic as well as the complete algorithmic design. We have also included a table of algorithmic hyperparameters to make the presentation more accurate and transparent.
> > > >
> > > > **Q3**
> > > >
> > > > Thanks for the patience. We train Transfuser on our preference dataset. Although Transfuser is not the strongest state-of-the-art planning method, its deployment is robust and provides a baseline performance. Since our encoder gets inspiration from Transfuser, it also provides a valuable ablation study on perference learning by TrajHF. We display the results in the table below.
> > > > |Test split|minADE|minFDE|Collision Rate↓|Offroad Rate↓|
> > > > |-|-|-|-|-|
> > > > |Aggressive|2.1377|5.4300|14.90%|6.21%|
> > > > |Defensive|1.1268|3.2822|8.66%|0.20%|
> > > >
> > > > Transfuser demonstrates a baseline and acceptable planning performance while significantly underperformes relative to our approach. This indicates the superiority of our diffusion generative design and RLHF alignment.
> > > >
> > > > **Q4**
> > > >
> > > > We appreciate the reviewer’s persistence on this point. Upon reflection, we agree that relying solely on static figures limits the assessment of dynamic behaviors, particularly when claiming distinct "aggressive" or "defensive" driving styles.
> > > > We have revisited our visualization pipeline and have prepared a short demo video from BEV to complement the quantitative analyses. We wish to clarify that our initial hesitation regarding "infeasibility" stemmed from the non-reactive preference dataset. However, we acknowledge that visualizing the open-loop planning evolution is entirely feasible and highly informative.
> > > >
> > > > We have included a video demonstration (available in supplementary material) that visualizes the qualitative cases discussed in Figure 4, detailed in Appendix B.5 in the revised manuscript version. The video highlights:
> > > >
> > > > ***Aggressive Overtaking:*** It visualizes the ego vehicle's decision-making process in the "opposing lane overtaking" scenario. The video demonstrates how the RL-finetuned policy consistently plans a lane-change trajectory to bypass static obstacles, whereas the pretrained model hesitates or follows behind.
> > > >
> > > > ***Defensive Yielding:*** It illustrates the "defensive" model proactively decelerating and yielding space when facing a cutting-in vehicle or ambiguous intersection, contrasting with the baseline's maintenance of nominal speed and collision risks.
> > > >
> > > > ***Temporal Consistency:*** The video displays the sequence of 4-second trajectory predictions over consecutive frames, offering a clearer view of the planner's stability than the static BEV snapshots in the main paper.
> > > >
> > > > We believe this visual evidence effectively substantiates our claims regarding style alignment and safety-related properties.
> > > >
> > > > **Q5**
> > > >
> > > > We thank the reviewer for emphasizing the importance of directly reflecting our discussions in the manuscript text. We fully agree that all clarifications and empirical nuances should be transparently documented in the paper itself. We have uploaded a revised version of the manuscript. The specific changes are detailed in the seperate comment session.

---

> > > > ### Author Response · Authors · 2025-11-29
> > > > **Response to Weakness**
> > > >
> > > > As required by the reviewer, we response directly and independently to the Weakness raised in the review.
> > > >
> > > > **1.Experimental results are not sufficiently strong to validate the main claims.**
> > > >
> > > >  We thank the reviewer for their careful examination of our experimental results and for highlighting the need for precise language regarding our performance on the NavSim benchmark. While we acknowledge that TrajHF does not surpass the absolute highest score currently on the leaderboard (GoalFlow), we emphasize the flexibility and multi-modality brought by our anchor-free diffusion generative ability, to support the downstream RLHF alignment. As we mentioned in the response to Reviewer KBTc Question 3 and Reviewer MMQc Question 1. The gap stems from our deliberate choice to avoid predefined anchors or vocabularies (as indicated in the "A/V" column of Table 1). We also provide a upper-bound seletion mechanism to demonstrate the optimal performance of our trajectory generative model. We prioritized our anchor-free design to support the continuous, flexible distributions required for our core contribution of personalized RLHF.
> > > >
> > > > **2.Personalization results rely entirely on private data.**
> > > >
> > > > We thank the reviewer for raising these critical points regarding reproducibility and the rigor of our evaluation. We acknowledge that the reliance on internal data and subjective metrics requires further transparency and statistical validation. We address these concerns below:
> > > > ***Data Availability and Quality*** We understand that utilizing an internal dataset limits direct reproducibility. To address this, we introduce a rigorous multi-stage pipeline described in Appendix A.1 to ensure high-quality annotations and reimplementation on any end-to-end dataset, including Scenario Mining, Key Frame Selection, Human Verification and Model Rejection.
> > > > ***The Role of Standard Metrics (ADE/FDE)*** We agree that ADE and FDE are insufficient for measuring style alignment. However, we retained them in Table 2 specifically to verify safety and feasibility rather than style. Additionally, we report Collision Rate and Offroad Rate as safety metrics. Our results show that RL finetuning improves style alignment (BOE) while improving safety compared to the pretrained baseline. This demonstrates that our method achieves personalization without catastrophic forgetting of general driving skills.
> > > >
> > > > ***Statistical Rigor of the BOE Metric*** We appreciate feedback regarding the statistical validation of the Better or Equal (BOE) metric. While we utilized 5 professional annotators to mitigate individual bias, we agree that variance and significance testing are necessary to confirm the results are not due to chance.
> > > >
> > > > ***3.Lack of safety and generalization analysis.***
> > > >
> > > > We thank the reviewer for highlighting the critical importance of safety verification, particularly when introducing "aggressive" driving styles. We report collision rate and offroad rate to address this concern. The offroad rate is shown below:
> > > > |Method|Collision Rate↓|Offroad Rate↓|
> > > > |-|-|-|
> > > > |w/o FT|12.50%|5.39%
> > > > |Aggressive RL|***2.70%***|***1.59%***|
> > > >
> > > > Our finetuned model show superior performance on planning safe trajectory in the complex emergent interaction scenarios.
> > > >
> > > > **4.Minor presentation and mathematical issues.**
> > > >
> > > > We thank the reviewer for their close reading of our manuscript and for identifying these presentation issues. We agree that rigorous mathematical notation and clear presentation are essential. We have carefully proofread the paper and incorporate the corrections in the revised manuscript. To clarify, $s_0$ indicates the initial state and is excluded from the trajectory prediction model in $x$ in Premilinary.

---

> ### Author Response · Authors · 2025-11-24
>
> **Q3:**
>
> ***1. Algorithmic contribution of DPGRPO. How does it differ from GRPO/DPO, and does it improve stability, sample efficiency, or alignment quality?***
>
> Our method builds on GRPO for diffusion policies but introduces two key modifications that are tailored to autonomous-driving trajectory planning (Alg. 1):
>
> ***(i) RL + BC joint objective for stable preference optimization.***
>
> DPGRPO keeps the original GRPO-style RL loss (Eq. 8–9), but additionally introduces a behavior-cloning (BC) loss on expert trajectories (Eq. 10), and optimizes their weighted sum (Eq. 11). This BC term is not present in existing GRPO implementations. In our setting with noisy, multi-objective preference rewards, the BC anchor substantially stabilizes training: it prevents the diffusion policy from drifting to unsafe or physically implausible behaviors while still allowing it to move away from the pre-training distribution to satisfy new driving-style preferences.
>
> ***(ii) Short supervised refresh to restore general-purpose driving behavior.***
>
> After DPGRPO training converges, line 12 in Algorithm 1 applies a short supervised “refresh” phase on the driving-style dataset. The RL phase encourages strong preference shifts and can make the policy overly aggressive or stylized in certain scenarios. The refresh step re-anchors the policy to fundamental driving constraints while preserving the preference-aligned behavior learned in the RL phase. This makes DPGRPO practically usable as a general-purpose planner across diverse scenarios, rather than a policy that over-fits to extreme preference signals.
>
> ***Contrast to DPO and empirical gains.***
>
> Unlike DPO, which directly optimizes preference likelihood and is highly sensitive to noisy or conflicting labels, DPGRPO uses a reward model and group-relative advantages, combined with BC regularization and the refresh phase, to handle multi-objective trade-offs more robustly. Quantitatively, across both aggressive and defensive driving styles, DPGRPO consistently outperforms the re-implemented PPO variant of Wang et al. (2024)(Table 5), a DPO baseline (Table 6), and a naïve GRPO baseline (As follow point 2.) on all trajectory-quality metrics. At the same training budget, DPGRPO achieves better alignment quality (lower miA/meA/miF/meF and higher Div) and avoids the training instability and failures we observe for DPO on the same preference dataset.
>
> ***2. A quantitative comparison or ablation isolating DPGRPO’s contribution***
>
> To isolate the contribution of the short supervised refresh phase, we compare full DPGRPO to a GRPO-only variant in which we remove short supervised refresh and train the policy using only the GRPO-style RL updates. The results are summarized in Table R1:
> |Method|Test Split|miA↓|meA↓|miF↓|meF↓|Div↑|
> |-|-|-|-|-|-|-|
> |Aggr. GRPO|Aggr.|2.79|6.87|8.08|19.14|0.67|
> |Aggr. GRPO|Normal|1.00|3.32|3.11|9.90|0.64|
> |Defe. GRPO|Defe.|0.84|3.38|1.78|8.05|0.51|
> |Defe. GRPO|Normal|0.63|4.29|1.07|9.93|0.49|
>
> Compared to the full DPGRPO model, the GRPO-only variant yields clearly worse preference-alignment metrics on both aggressive and defensive preference splits, and it also reduces trajectory diversity (Div). This confirms that short supervised refresh is not a cosmetic implementation detail but a crucial component of DPGRPO: the RL phase alone tends to produce overly stylized, sometimes excessively aggressive trajectories, whereas the refresh phase restores general-purpose driving behavior and recovers performance on standard test sets while maintaining strong alignment to the target driving style.
>
> **Q4: Evaluation of personalization on NavSim? Provide results of other state-of-the-art methods on the internal preference datasets for a fairer comparison?**
>
> We appreciate the reviewer’s suggestion. While NavSim is an excellent benchmark for general driving capabilities, its metrics (PDMS) primarily evaluate feasibility, progress, and comfort based on ***average*** human driving standards. It does not contain ground truth labels for specific driving styles (e.g., "aggressive" or "defensive"). Consequently, a model successfully conditioned on an "aggressive" style (characterized by higher jerk or closer following distances) might inherently receive a lower score on NavSim's "Comfort" metric. Therefore, comparing conditioned vs. non-conditioned models on NavSim mainly validates that personalization does not catastrophically degrade general safety, but it cannot effectively quantify alignment with specific driving styles. We have conducted the comparison between the non-conditioned (Base) and preference-conditioned (Finetuned) models. To provide a fairer comparison and demonstrate alignment improvements, we are evaluating a representative state-of-the-art method, Transfuser, on our Internal Preference datasets. Because of the time constraint, we cannot report all the results currently. We will add the metric results in comments as soon as possible and revise the paper simultaneously.

---

> ### Author Response · Authors · 2025-11-24
>
> **Q5: Safety considerations: How do the authors ensure that finetuning toward “aggressive” preferences does not violate safety constraints or produce unsafe behavior? Are there empirical checks, such as collision rates or safety rule compliance?**
>
> We thank the reviewer for highlighting the critical aspect of safety. We explicitly monitored the Collision Rate to ensure that finetuning toward "aggressive" preferences does not compromise safety. We conducted a safety evaluation on the style-specific test splits. The results, summarized in the table below, demonstrate a significant improvement in safety for the aggressive style.
>
> |Method|Collision Rate↓|
> |-|-|
> |Base Model|12.50%|
> |TrajHF (Finetuned)|2.70%|
>
> Contrary to the concern that aggressive driving might be unsafe, our results show that the collision rate on the "Aggressive" split dropped significantly from 12.50% to 2.70%.This is because the "Aggressive" dataset consists of complex interaction scenarios (e.g., overtaking, cut-ins) mined from human takeover data. In these high-stakes situations, the Base Model often behaves hesitantly or fails to complete the maneuver in time, leading to failures. By finetuning, TrajHF learns to be assertive and decisive—aligning with the expert human strategy—thereby navigating these interactions more safely and effectively.
> It is worth noting that the absolute collision rates reported above are influenced by inherent noise in the raw dataset, particularly inaccuracies in the ground truth yaw (heading) annotations, which makes precise collision calculation challenging even for ground truth trajectories. Therefore, we emphasize the relative trend over the absolute values.
>
> **Q6: Details on PPO and DPO variants: In Appendix C.1, the paper reports results using PPO and DPO variants. Could the authors provide more implementation details on the PPO setup, specifically, the design of the critic network and reward model within that framework?**
>
> ***1. Details on PPO and DPO variants.***
>
> The following Table  summarizes the hyperparameters used for DPO and our PPO variant:
> |Parameter|DPO variant|PPO variant (Wang et al. 2024)|
> |-|-|-|
> |Batch size|32|32|
> |K sample|8|8|
> |Learning rate|5e−5|5e−5|
> |Max epoch|30|30|
> |Epsilon Cliping|None|0.2|
> |KL coef|None|0.01|
> |Inverse temperature|200|None|
>
> Regarding the PPO setup, we follow the overall framework described in Wang et al. (2024), but we do not directly reuse their hyperparameters. There are two main reasons for this. First, the tasks are not identical: Wang et al. focus on lane-changing, while our work addresses general trajectory prediction, which involves different input distributions and behavioral variability, leading to different training dynamics. Second, the model architectures differ substantially—Wang et al. use an LSTM-based model, while we employ a diffusion model. These architectures exhibit different gradient scales, stability characteristics, and convergence behaviors, making it inappropriate to directly transfer their hyperparameter settings such as learning rate, batch size, or training schedule.
> To ensure a fair and controlled comparison, we align the key optimization hyperparameters of PPO (e.g., learning rate, batch size) with those used in our primary baseline DPGRPO. This strategy avoids confounding factors arising from architectural and task differences while maintaining comparability across methods.
>
> ***2. The design of the critic network and reward model within that framework?***
>
> For DPO, training is performed directly from pairwise preference signals without using an explicit critic or reward model.
> In our PPO variant, we do not use a separate critic network. This decision is based on two considerations. First, our PPO implementation aims to reproduce the preference-finetuning setup described in Wang et al. (2024), but the reference work does not release code nor specify any details about the critic architecture.  Second, our task involves a single-shot generation with only a sequence-level terminal reward. In this structure, a critic’s temporal-difference value estimation provides little benefit. We instead rely on advantage normalization to achieve stable variance control.
> Regarding reward design, we follow the descriptions in Wang et al. (2024) and adapt them to our task by constructing a hybrid reward to guide PPO updates. This reward consists of two components: (i) an MSE-based trajectory consistency term, and (ii) a style/quality score produced by a preference-trained reward model, which is also used in our DPGRPO algorithm. During PPO training, this hybrid reward is combined with the standard clipped PPO objective and a KL regularization term to ensure stable and comparable optimization.

---

### Author Response · Authors · 2025-11-29
**Paper Revision**

Thanks for the valuable suggestions and comments from all the reviewers, which help a lot for our paper revision. We decide to integrate all the additional experiments and clarification into a new revision of our paper. The modification includes:

***1. Clarification of performance on NavSim benchmark:*** we use "competitative" instead of "comparable" following the advice from Reviewer n2yq.

***2. Additional experiments:*** we add seperate validation of reward model in Appendix A.3 required by Reviewer MMQc Question, ablation of short SFT refresh in Appendix C.4 following advice from  Reviewer ny2q Question3, collision rate and offroad rate on "aggressive" preference dataset and comparison with Transfuser on preference dataset in Appendix B.7 suggested by Reviewer ny2q Question 4.

***3. Clarification of Wang et al. (2024) variant:*** thanks for the careful observation from Reviewer ny2q Question 6. We add parameters comparison  between our TrajHF and Wang et al. (2024) in Appendix B.4. Also, we replace all instances of “PPO variant” with “Wang et al. (2024) variant”.

***4. Add BEV video visualization:*** as required by Reviewer ny2q Question 7, we add BEV video visualizations in supplementary material and describe in Appendix B.4 in detail.

***5.Minor presentation and mathematical issues:*** we correct the presentation issues mentioned by Reviewer ny2q.

---

### Meta-Review · Area_Chair_KkZp · 2026-01-05

**Summary:**

This submission proposes TrajHF, a framework for personalized autonomous driving trajectory planning by combining (i) a diffusion-based trajectory policy with a multi-conditional denoiser conditioned on multi-modal inputs (camera/LiDAR/action history), and (ii) a preference alignment finetuning stage using RLHF. The finetuning algorithm, DPGRPO, adapts group-relative advantage optimization (GRPO) to diffusion policies and introduces stabilization via BC regularization and a short supervised refresh phase. Experiments include the public NavSim benchmark and internal preference datasets involving “aggressive” vs. “defensive” driving styles, evaluated using standard trajectory metrics plus a human preference score (BOE).

Several critical concerns raised by the reviewers are: (a) lack of SOTA results on the public benchmark, (b) reliance on private datasets for the core personalization evaluation, and (c) insufficient safety/generalization analysis and limited qualitative validation. The rebuttal has addressed the concerns to some degree, however, the major issues remain unresolved: (1) lack of strong public benchmark improvements, (2) reliance on private data for the central personalization claim, and (3) incomplete safety/generalization evaluation.

**Reviewer Concerns:**

A) Benchmark performance: Multiple reviewers object to the claim that TrajHF is “comparable to state-of-the-art” on NavSim because it is below top methods like GoalFlow.
* ny2q: NavSim performance is “competitive but not SOTA,” contradicting the paper’s phrasing; requests that language be corrected.
* KBTc: Performance is below GoalFlow and this weakens public evidence.
* MMQc: Reiterates that SOTA is not achieved on navtest and abstract claim should be toned down.
Rebuttal response: Authors agree to revise “comparable” → “competitive,” and emphasize that oracle PDMS selector shows an upper bound (94.3) but deployable EM is lower.

B) Reliance on private internal datasets for personalization + weak baseline comparisons: The main personalization claim is validated mostly on private preference data. Reviewers want stronger reproducibility and comparisons.
* ny2q: The internal dataset is not public; worries about annotation quality, bias, label balance; wants baselines evaluated on internal preference data.
* KBTc: “Key wins rely on internal datasets, semi-synthetic pairs, and BOE.”
* MMQc: Generally accepts internal evaluation but notes some simplification of “aggressive vs defensive.”
Rebuttal response: Authors cannot release dataset, but promise pipeline + possible partial release (sanitized annotations / reward models). They add a comparison with Transfuser on preference data (but admit it’s not the strongest SOTA baseline). Reviewer ny2q still notes that more comparisons remain important.

C) Safety & generalization concerns: Personalization could produce unsafe behaviors (especially “aggressive”), but reviewers felt safety evidence was initially insufficient.
* ny2q: Requests collision/rule compliance; distribution shift robustness.
* MMQc: Safety feasibility claim not well supported; aggressive may be accident-prone.
* KBTc: Asks about TTC and comfort after style tuning.
Rebuttal response: Authors report collision rate and offroad rate on aggressive split, showing large collision reduction after finetuning (12.5% → 2.7%) and improved offroad. They argue aggressive split contains complex interaction scenarios where decisiveness improves safety.
Remaining weakness: They largely argue that TTC/comfort metrics conflict with “aggressive” style; this may not fully satisfy reviewers wanting principled safety evaluation or generalization across distributions.

**Reviewer Scores:**

4,4,6. The review scores likely remain the same.

---

### Decision · Program_Chairs · 2026-01-26

Reject